



# SMC-Floods database: A high resolution press database on floods for the Spanish Mediterranean Coast (1960-2015).

Salvador Gil-Guirado[1,2], Alfredo Pérez-Morales[2], Francisco Lopez-Martinez[2]

[1]Interuniversity Institute of Geography, University of Alicante, P.O. Box 99, 03080 Alicante, Spain.
[2]Department of Geography, University of Murcia, Campus de la Merced, 30001 Murcia, Spain

*Correspondence to*: Salvador Gil-Guirado (salvagil.guirado@ua.es)

**Abstract.** Floods are the natural disaster that affects the greatest number of people and causes the highest economic losses in the world. However, some areas, such as the Mediterranean Coast of the Iberian Peninsula, are especially exposed to this natural hazard. The problem takes on even more relevance when a changing social dynamic is added to the natural context.
With a view to accomplishing correct spatial planning in the light of the flood hazard, it is necessary to carry out an exhaustive analysis of the spatiotemporal variability of floods with a scale of analysis that allows the detection of changes and the search for causality. Databases compiled from journalistic documentation offer these possibilities of analysis and represent a vital tool for correct spatial planning.

In this study we present the SMC (Spanish Mediterranean Coast)-Flood Database for the municipalities of the Mediterranean
coast of mainland Spain. This database has enabled the reconstruction of 3,008 cases of flooding on a municipal scale and with daily resolution, with information on the type of damages, intensity and area affected. The spatiotemporal analysis of the data reveals black spots where floods are especially intense and damaging, compared to highly-developed areas where the frequency of the floods is very high. This situation is especially worrying, insofar as we have detected a growing trend in the frequency and area affected by floods. However, it is positive that the intensity and severity of the floods follows a falling trend. The
main novelty lies in the fact that the high-resolution spatial analysis has made it possible to detect a clear latitudinal gradient of growing intensity and severity with a north-south direction. This pattern subjects the coastal municipalities of the south of Spain to a complicated adaptation scenario.

**Keywords:** SMC-Flood database, newspaper, Spanish Mediterranean Coast, flood intensity, flood severity

## 1 Introduction

On the Spanish Mediterranean coast, the relationship between water systems and societies has been marked over time by the succession of periods of drought and catastrophic floods. This dual system has determined the exposure of societies to the reception of flows for agricultural and domestic use, in an environment characterized by the torrential nature of rainfall (Gil-Guirado, 2013). Together with the climatic situation it is also necessary to consider the social component. The situation of economic growth experienced on the Spanish Mediterranean coast in recent decades has generated an inordinate increase in
exposure to the hazard (Pérez-Morales et al., 2018) with a significant rise in economic losses caused by floods (Barredo, et al.,





2012). This growth process has occurred without proper planning to reduce the impact of floods (Olcina-Cantos et al., 2010). One of the factors that facilitate the lack of strategic planning is the absence of a correct chronology of flood episodes (Hilker et al., 2009). In this situation, land use plans are based on inadequate chronologies that do not report the real risk of the population of this area (Barriendos et al., 2014).

Several open and global flood databases have been developed over recent years by different bodies such as universities (Brakenridge, 2010; EM-DAT, 2018), insurance groups (Munich Re, 2017; Swiss Re, 2018), climatic agencies (WMO and UCL, 2014; NOOA, 2018; ESSL, 2018) or international organizations (Rodier and Roche, 1984), the results of which have boosted flood risk knowledge on a global and regional scale. One of the databases that include the best spatial coverage of floods is the European Past Floods dataset of the European Environment Agency. This database contains information on past

floods in Europe since 1980 (European Environment Agency, 2018) including the impact type and losses. However, the database has two limitations: i) the level of spatial resolution is variable between the municipal and regional scales, and ii) there is a considerable underestimation of the number of events owing to the use of indirect sources (Llasat et al., 2013a). The studies of Adhikari, et al. (2010), Bouwer (2011), Llasat et al. (2013a) and Napolitanao et al. (2018) include a detailed catalogue of some of these databases and their scope. Furthermore, their results have been used by a great deal of research to

analyse the trends and changes observed in the behavior of floods on different scales. For example, Barredo (2007 and 2009) analysed the losses caused by floods for Europe between 1950 and 2006. Similarly, Kundzewicz et al. (2013) used the flood database of the Dartmouth Flood Observatory to analyse the trends of severity and scale of the floods in Europe between 1985 and 2009. Other types of studies have looked into the spatiotemporal reconstruction of fatalities due to flooding in the USA. For example, Terti et al. (2017) for the 1996 to 2014 period and Ashley and Ashley (2008) for 1959 to 2005. Neumayer and

Plümper (2007) have analysed the impact of floods on the life expectancy of women on a national scale for a set of 141 countries between 1981 and 2002. Other studies such as that of Adhikari et al. (2010) have integrated the information of each flood event existing in the main databases on a global scale to increase the spatial resolution and improve the spatiotemporal representativeness of the data. Finally, Jongman et al. (2015) examine the role of vulnerability and capacity for adaptation of societies to confront floods between 1980 and 2010, to conclude that there is a trend for convergence in vulnerability levels

between low- and high-income countries. However, the results of these studies may present biases relating to the underestimation of the number of episodes and the failure to consider local social-environmental variations. In addition, there are other problems related to the dispersion of the type and quality of information related to the same flood event (Guha-Sapir and Below, 2002) as well as the existence of inclusion criteria based on the crossing of human or economic loss thresholds (Hirabayashi et al., 2008).

On the other hand, there are several works like this that have developed their own databases according to primary sources (newspapers, books, reports and technical and scientific papers) for different regions of the planet. With regard to the classification of historic sources, according to their originality and the way they have been generated, these may be (Barriendos et al., 2014): Primary or direct, when the material is first hand in relationship with the fact to be studied, that is, prepared at the same time as the event and by a direct witness; secondary if they are based on the primary sources; and tertiary, those made



on the basis of the previous two. Secondary and tertiary sources are indirect sources. In this respect the Mediterranean countries stand out. Some works have made a remarkably effort to synthesize the different European regions results ir order to offer a homogeneous database for the western Mediterranean. In this regard, the works of Llasat et al. (2013a, 2013b) show the FLOODHYMEX database results (produced in the framework of the HYMEX Project). These works collect a large amount
of data on flood events occurred between 1981 and 2010 trough newspaper sources with a high spatial resolution. As for Spain, are remarkable the results obtained by Llasat et al. (2009; 2013a, 2013b; 2016) and Barnolas and Llasat, (2007) through the INUNGAMA floods database. This database completes another high-resolution database that covers the north-east of Spain between 1982 and 2007 and also based on newspaper sources. Also for Spain, but for a much longer period of time (1035-2013), Barriendos et al. (2014) make use of another database obtained from historic documentation, newspapers, official
reports and studies by experts. Also in the Iberian Peninsula, Zêzere et al. (2014) present a database for Portugal and for the period 1865 to 2010, obtained via 16 national, regional and local newspapers. Other Mediterranean countries present equally valid initiatives, such as Diakakis et al. (2012) for Greece, whose work covers the period from 1880 to 2010 via the use of journalistic sources and flood event databases from state civil protection agencies. Italy also has adopted a large amount of initiatives to ascertain the flood risk of its populations with a high level of spatiotemporal resolution. In this respect, projects
have been developed for specific regions of Italy. Specifically, in the region of Calabria, Polemio and Petrucci (2012) and y Petrucci et al. (2018) analyse the variability of the floods at municipal level between 1880 and 2007 using for their reconstruction journalistic information and historic documentation. For the region of Campania, Vennari et al. (2016) reconstruct more than 500 flood events for the 1540 to 2015 period using for this purpose historic documentation of different types. However, the project with the largest scale, the AVI Project, was developed on the basis of the studies of Guzzetti et al.
(2005) and Salvati et al. (2013), whose results enabled the establishment of a high-resolution floods and landslides database for the whole of Italy between AD 68 and 2010. Basically, the AVI Project uses primary documentation, but with detailed information on fatal victims, people and displaced persons.

Outside the Mediterranean area, other regions with a high risk of flooding have conducted in-depth studies of this type. For Australia, FitzGerald et al. (2010) construct a database of people killed by floods between 1997 and 2008 by means of data
from newspapers and historic accounts, as well as government and scientific reports. In addition, there are notable initiatives for the compilation of flood events for long periods of time via historic documentary sources. For example, McEwen (2006) for Scotland between 1200 and 2004; Glaser and Stangl (2004) for Central Europe between 1000 and 2003; Quan (2014) for Shanghai 251 to 2000; and Brázdil et al. (2014) for Southern Moravia (Czech Republic) between 1650 and 2000. Without doubt, these studies represent a notable advance in research on flood risk in their respective areas of study.
In short, the majority of all these works is related to analysing the flood trends for the period of time reconstructed. However, the relationship between the increase in exposure, losses and impact of the floods does not follow a growing lineal function. In fact, climate variability, defence infrastructures, adaptation measures and the increase in exposure have changed both social perception and flood trends (Jongman et al., 2014). In this respect, it is necessary to stress the existence of a negative correlation between the duration and the direction of the trends, whilst the negative trends appear in the studies with data relating to several



centuries, the positive trends appear in studies that analyse data of the last half-century. These divergences are due both to the capturing of climatic oscillations in data with a long duration (Barriendos et al., 2014) and to the heterogeneity of the sources used during recent years (Brázdil et al., 2014). Furthermore, the increase in exposure to flood risk has led to a rise in the trends (Perez-Morales et al., 2018).

Among the different sources used for flood databases, journalistic sources allow a homogenization of the documentary volume of different countries during at least the last 150 years. In fact, the majority of the studies referred to which reconstruct floods for more recent periods have found their main source of data in newspapers (e.g. FitzGerald et al., 2010; Zêzere et al., 2014). Despite the fact that journalistic sources describe the impacts of floods on societies in great detail, their high spatiotemporal resolution and the quantity of information dealt with make it necessary to reduce both the area of study and the period analysed.

Obviously, although the compilation of information is an arduous task with detailed archive work, but its results reflect the impact of the floods accurately (Barriendos et al., 2014).

To reduce the knowledge deficit regarding the spatiotemporal variability of floods and contribute to a more efficient zoning of the Mediterranean coast according to flood risk, we have developed a high-resolution flood database. This database, called SMC-Flood, includes all the floods cases recorded in newspapers for the different municipalities of the Spanish Mediterranean

coast from 1960 until 2015. In this regard it must be clarify the different between flood cases and flood events. We consider a flood case when a municipality has suffered some economic or social impact due to rain on a specific day. However, a flood event refers to an atmospheric situation during a specific day or time period that may have affected several municipalities at the same time (several flood cases). In this way, a flood case always corresponds to an affected municipality on a particular day, while an event may involve several municipalities and days. For example, during the flood event of October 1973, there

were 27 flood cases over the coast of the provinces of Almeria and Murcia between the 18[th] and the 19[th] of that month. The fact of considering cases and not events motivates the large number of records against databases that only consider flood events. However, the aim of this paper is to present the cases at the local level as an indicator of the research potential of this type of high resolution databases.

The methodology consists of exploring the archives of the main newspapers published in the area. The searches have been

made by typing the name of each of the 179 coastal municipalities followed by 7 key terms. Later, each flood was classified according to dates, intensity level and damage type. Finally, we consulted the specific bibliography to rule out any data gaps. This database is expected to be used for various purposes, such as the evaluation of flood prediction tools, the determination of acceptable risk thresholds according to the conditions of exposure and vulnerability, the characterization of seasonal and regional trends and, in general, the SMC-Flood database will contribute to improving the understanding of the flood processes

in an area of special economic, climatic and social interest.

The article is structured in four parts. First, the study area is explained and justified. The second part sets out the methodology used for the compilation of the SMC-Flood database. The third part presents the results obtained from the preliminary exploitation of the data. This is based on a general summary of the database, a spatial analysis of the distribution of flood episodes, a time analysis of when such events took place, an exposé of the variability of the damages caused by those floods





and, the evolution and trends are presented for the period analysed. Finally, the main conclusions are presented followed by the lines of work to be followed.

## 2 The Spanish Mediterranean Coast: "a flood risk-region"

The study area includes all the Coastal Municipalities of the Spanish Mediterranean Sea on the Iberian Peninsula. In total there are 179 municipalities integrated in 11 provinces and 4 Autonomous Communities (See Fig. 1). The total area is 13,381 Km$^2$ (2.64 % of total Spain area) with a population of 8,413,290 inhabitants in the year 2016 (18% of the Spanish population) (INE, 2018) and an average population density of 1,200 inhab/km$^2$, a figure far higher than the average for the EU (119) and for Spain (92).

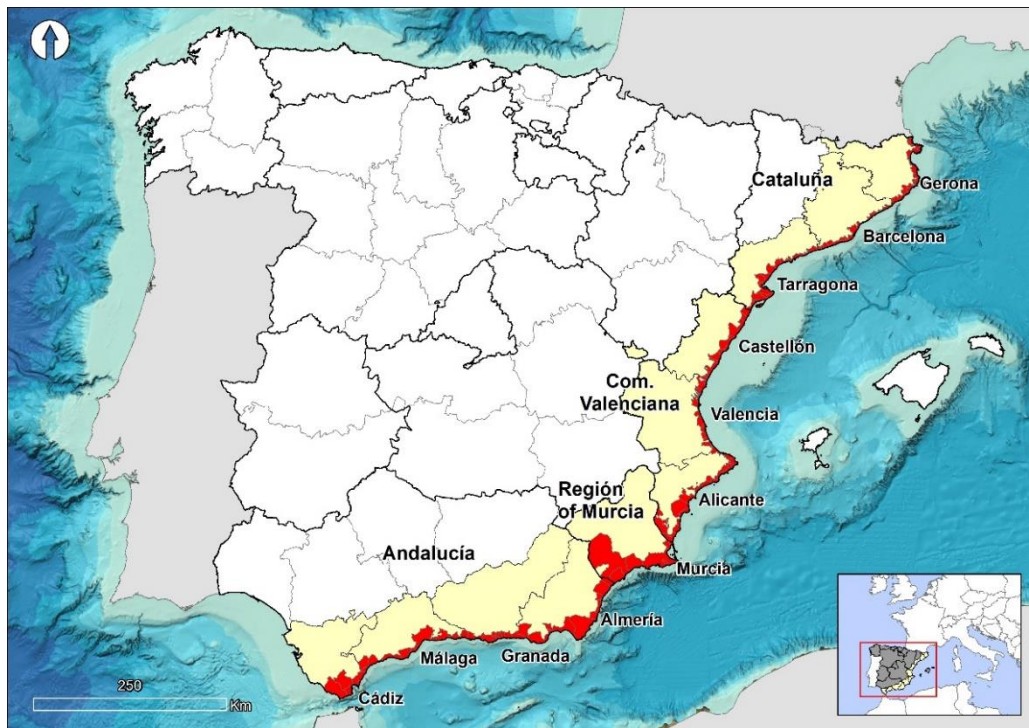

**Figure 1:** Coastal Municipalities of the Spanish Mediterranean Sea over the Iberian Peninsula.

The climatic and hydrological conditions of the Spanish Mediterranean basins, together with the intensive anthropic transformation that has taken place, have converted this space into a "risk region" with a high level of vulnerability (Olcina-Cantos et al., 2010). The rainfall climatology in the western Mediterranean is marked by high variability coefficients (above 35%). Thus, 25% of rain days concentrate more than 75% of precipitation (Martín-Vide, 2004). Seasonality of torrential rains over the Spanish Mediterranean region is marked by a maximum at the end of summer and especially during the autumn (Llasat et al., 2013a). This maximum is due to warm, humid air coming in at low levels from the sea (Gilabert and Llasat, 2018: 1864). That atmospheric situations can be emphasize with the presence of a closed upper-level low (Sumner et al., 2003), which has





become completely displaced (cut off) from the basic current and current independently of that current (Pagán et al., 2016). However, convective precipitations are the trigger for a large number of torrential rain episodes of low spatial extent and especially related to flash floods (Gilabert and Llasat, 2018). This climate scenario can become more dramatic in the future. Sumner et al. (2003: 800) pointed out an important increase in most of the synoptic situations with an easterly flow in the

Spanish Mediterranean coast. This situations are prone to generate torrential rains and a flood hazard increase.

Added to this climatic scenario is the effect of an abrupt relief, with sharp gradients (Gilabert and Llasat, 2018) and scarce vegetation, which increases the quantity of effective rain converted into run-off. Furthermore, the presence of pre-littoral reliefs exacerbates these precipitations and explains part of the great spatial variability of the precipitation during a single atmospheric event (Romero et al, 2000).

In addition, this environmental scenario is complicated with the addition of the social component. Intensive agriculture, industry in the major urban centers, trade and tourism make this region the main center of urban growth (Burriel, 2015), economic dynamism and with one of the highest rates of population and economic growth of Europe in the last 50 years. This process was boosted in the early nineteen-sixties, when Franco's regime started to be more open abroad. Furthermore, climate conditions (warmer temperatures and a large number of sunny days) played a major role in tourist arrivals, in fact it became

the slogan of the area (Cortés-Jiménez, 2008).

## 3 Methodology and sources

The SMC-Flood database contains information about floods cases at municipal level published in printed newspapers and which took place in the Spanish Mediterranean Coastal Municipalities (henceforth SMCM) between AD 1960 and 2015. The newspapers used were selected according to the following criteria: i) it has the highest circulation in each one of the four

Autonomous Communities studied, and ii) it has its head office in the same Autonomous Community (See Table 1). This criterion ensures the reliability of the data, since news coverage of the flood is more extensive when the source of origin of the data is a newspaper that has its office in the same Autonomous Community.

**Table 1:** Newspaper sources for the MEDIFLOOD-Database:

| Newspaper | Type of Access | Newspaper Library Link | Period | Headquarters |
|---|---|---|---|---|
| ABC | Open | http://hemeroteca.abc.es/avanzada.stm | 1903-Now | Andalucía |
| LV | Open | https://www.lavanguardia.com/hemeroteca | 1881-Now | Cataluña |
| EMV | Restricted | - | 1872-Now | Comunidad Valenciana |
| LVM | Restricted for news before 2006 | - | 1903-Now | Región de Murcia |

*The newspaper ABC has its head office in the city of Seville (Andalusia), the newspaper La Vanguardia (LV) has its head office in the city
of Barcelona (Catalonia), the newspaper El Mercantil Valenciano (EMV) has its head office in the city of Valencia (Valencian Community). Finally, the newspaper La Verdad de Murcia (LVM) has its head office in the city of Murcia (Region of Murcia.

The information from these newspaper archives is available digitally, both with open access (La Vanguardia and ABC), and restricted access (Levante Mercantil and La Verdad de Murcia). When access was restricted, we obtained an unrestricted password for the Levante-EMV, obtained free of charge within a scientific cooperation framework. In the case of La Verdad

de Murcia we carried out the search on the central computer of the head office of the newspaper in the city of Murcia.



The digitization of documents enabled the information search to be performed by keywords in the search engines of the archives of each newspaper. In this respect, the first step consisted of relating each municipality (179) with the newspaper corresponding to its Autonomous Community. However, the search for information in any newspaper served in some cases to complete the level of detail on specific cases and municipalities. Additionally, we consulted the specific bibliography to rule

out any data gaps[1]Taking into account the problems related to the use of newspaper sources, such as: inhomogeneity, duplicity of information or contradictory information (widely discussed by Llasat et al., 2009 and 2013a), it makes especially necessary the results to be validate. Nevertheless, these newspaper source problems are more evident as we go back in time (more than 50 years), since the consistency of journalistic sources is greater in recent decades. Dispite de above, the fact that the query procedure of each news item has been done manually by the authors, ensures that some of the indicated problems are not

present in this database.Secondly, we carried out the systematic search for news where the name of each municipality appears together with any of the 7 keywords/phrases selected (See Fig. 2 Panel a). These keywords correspond to the most common ways of referring to a situation that is likely to describe a flood in Spain:

1. "Inundación" (Flood).
2. "Inundaciones" (Floods).
3. "Riada" (Flash flood).
4. "Lluvias torrenciales (Torrential rains).
5. "Fuertes lluvias" (Heavy rains).
6. "Intensas lluvias" (Strong rains).
7. "Tromba de agua" (Severe downpour).

In addition, the search was duplicated for those municipalities in Catalonia and the Valencian Community that have language variations (e.g. Alicante/Alacant, La Escala/L'Escala, El Puerto de la Selva/El Port de la Selva, Sagunto/Sagunt). In this way, we ensured that the same search criteria were complied with in all cases.

This initial search produced more than 1,500,000 possible results (news pages). Obviously, several keywords may be present on one page of news. In many other cases, the name of a municipality may appear on the page of a news item on floods, but

without having been affected by that flood. In relationship with this last point, it is possible to limit the search to cases in which the keywords are directly connected to the name of the town (i.e. "Torrential rains in Barcelona"). But we found that in this way the results were excessively restricted, and many news that did report on a flood were missing, since it is frequent for the keywords to appear in the headlines and for the body of the news item to describe the impacts and define the municipalities affected.

To filter the initial search results, each news item was saved in a digital file with the date of the news, followed by the initials of the newspaper (LVG, ABC, LVM or EMV) and, finally, the page of the news (page of the newspaper where the news was located). On many occasions, news about a flood in a specific town is included on the same day on different pages, offering varied and complementary information. Fortunately, this system for the coding of news enabled the elimination of duplications

---

[1] Basically, the National Catalogue of Historic Floods (Catálogo Nacional de Inundaciones Históricas) (Pascual and Bustamante, 2011): http://www.proteccioncivil.es/cnih



in both the keywords and the municipalities (the same piece of news describes floods in various municipalities). Thus, the file of possible floods was reduced to 23,580 pieces of news for the SMCM.

In the next step, qualitative information was transformed to quantitative information (See Fig. 2 Panel a). To this end we proceeded to code the text of the news in spreadsheets based on the consultation of all the news filed. The digitization of the

news sheets by means of Optical Character Recognition (OCR) substantially facilitated this arduous task. In turn, the coding complied with the following classification protocol (See Fig. 2 Panel b): On the one hand, every flood was assigned its exact date of occurrence (the date of the flood is at least one day before the date of the news). Next, the affected municipality or municipalities were defined. Finally, the intensity of each flood was determined according to 3 levels (Camuffo and Enzi, 1996; Barriendos et al., 2003; Llasat, et al., 2005; Barriendos et al., 2014):

- Level 1 (L1): **ORDINARY** flood. A flood without overflow and minor damages.
- Level 2 (L2): **EXTRAORDINARY** flood. A flood with overflow and major damages.
- Level 3 (L3): **CATASTROPHIC** flood. A flood with overflow, general destruction and deaths.

Level 1 floods not only consider the cases that have caused an ordinary flood in the river flow, but also take into account flash floods and in situ floods outside the rivers floodplain. For this reason, Level 1 floods are valid to report the climate (changes

in rain patterns) and social (changes in exposure or vulnerability) variability (Llasat et al., 2016), but they are not valid to report the hydrological rivers variability. Accordingly, some works that analyze the hydrological variability of the rivers exclude the L1 floods of their analyses (Llasat et al., 2005). In addition, we obtained 10 dichotomic variables to point to the presence (1) or absence (0) of any of the following effects/damages produced by the flood in each municipality. Thus, the information regarding the type of damage suffered was categorized in a simple manner, being aware of the difficulty involved

in objectivizing quantitative information consistently in time and space (Gil-Guirado et al., 2016). The variables created allow an approximate idea of the scope of the damages of each flood:

- Agriculture.
- Cattle.
- Fishing.
- Roads.
- Industry.
- Trade.
- Buildings.
- Tourism.
- Fatalities.
- Injured.



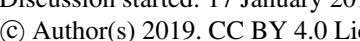



**Figure 2:** Method of cataloguing news step by step (**Panel a**) and example of the coding system for the news (**Panel b**). Source: in Panel b: ABC Newspaper, news of 23 August 2007.

Finally, with the results of the SMC-Floods database, we have conducted a trend analysis to analyse if floods and their intensity
5  have increased or decreased over time. To determine whether or not statistically significant trends exist, the improved non-parametric test of Hirsch and Slack (1984) has been used, based on the Mann-Kendall range widely used in climatic and



hydrological studies. This test informs on two possible hypotheses: the null hypothesis (H0) which defends that the series does not present a significant trend and the alternative hypothesis (Ha) which informs of a statistically significant trend, which may be negative or positive. The level of significance chosen has been 95%. In addition, Sen´s Slope has been calculated, which informs on the bias and size of this trend, multiplying this value by the total number of observations, we would obtain an approximate value of the mean loss or gain of the variable over the time period.

## 4 Results

### 4.1 SMC-Flood records summary

According to the SMC-Flood database, between 1960 and the year 2015 the SMCM suffered 3,608 floods cases. Of these, 72% were of an ordinary intensity (Level 1), less than 25% were extraordinary (Level 2) and slightly more than 3% catastrophic (Level 3) (See Table 2 Panel a).

With regard to the type of damages (See Table 2 Panel a), roads (almost 80%) and homes (45%) were the variables most affected by the floods. Trade and agriculture are also sectors that repeatedly suffer damages (in approximately 20% of cases). Tourism is another sector that suffers the impact of floods (16%). However, there are important differences in the type of damages according to the level of intensity. In general, the amount of damages increases according to the level of intensity. That is, the greater the intensity, the more assets and people are affected. Between intensity levels 1 and 2, the largest increases occur in residential properties (almost 80% of the level 2 floods involve damages to homes) and trade (almost half of level 2 floods involve impacts on trade). These two types of damage are interrelated, insofar as the overflow of the water body that affects residential properties also affects trading establishments, generally located in the lower part of the buildings. Also notable are the increases in the impacts on agriculture and trade. Regarding the changes between the damages produced in level 3 floods and those of other levels, the most notable is the increase in the direct effects on the health of people (fatalities and injured). In fact, almost 100% of cases of level 3 involve human victims. Indeed, the very classification method includes in its main criteria that, if a flood causes victims, this is a fundamental factor to be considered Level 3. In general, catastrophic floods are characterized by the fact that they affect all the economic and social fabric.

Furthermore, some ancillary indices and variables have been calculated to characterize floods in the SMCM (See Table 2 Panel b). The Severity Index is the sum of the quantity of damages multiplied by the intensity of each flood case. With regard to the average area affected, we have assigned to each flood the area of the municipality where it took place. While we are aware that a flood does not affect the whole of an administrative area, we consider that it is a good measure to perform comparative analyses on a spatiotemporal scale. Moreover, although a flood does not have a direct impact on the whole municipality, the effects are felt indirectly throughout the administrative territory. This calculation informs that, in the SMCM, each flood affects an area of 119 km$^2$. However, this value rises alarmingly as the intensity level of the floods increases.



**Table 2:** SMC-Flood Database summary.

| | | L1 | | L2 | | L3 | | TOTAL FLOODS | |
|---|---|---|---|---|---|---|---|---|---|
| | | N | % | N | % | N | % | N | % |
| | Cases | 2,599 | 72.03 | 887 | 24.58 | 122 | 3,38 | 3.608 | 100 |
| | Agriculture. | 346 | 13.31 | 307 | 34.61 | 49 | 40.16 | 702 | 19.46 |
| | Cattle. | 10 | 0.38 | 33 | 3.72 | 13 | 10.66 | 56 | 1.55 |
| | Fishing. | 46 | 1.77 | 70 | 7.89 | 5 | 4.1 | 121 | 3.35 |
| **A)** | Roads. | 1,928 | 74.18 | 807 | 90.98 | 108 | 88.52 | 2,843 | 78.8 |
| | Industry. | 30 | 1.15 | 84 | 9.47 | 24 | 19.67 | 138 | 3.82 |
| | Trade. | 225 | 8.66 | 432 | 48.7 | 61 | 50 | 718 | 19.9 |
| | Buildings. | 850 | 32.7 | 697 | 78.58 | 82 | 67.21 | 1,629 | 45.15 |
| | Tourism. | 298 | 11.47 | 239 | 26.94 | 37 | 30.33 | 574 | 15.91 |
| | Fatalities. | 10 | 0.38 | 9 | 1.01 | 118 | 96.72 | 137 | 3.8 |
| | Injured. | 27 | 1.04 | 65 | 7.33 | 45 | 36.89 | 137 | 3.8 |
| | | L1 | | L2 | | L3 | | TOTAL FLOODS | |
| **B)** | Severity Index | | 1.45 | | 6.18 | | 13.33 | | 2.57 |
| | Area (Km$^2$) | | 115.86 | | 118.04 | | 181.48 | | 118.62 |

*The different colours represent how the values deviate, above (red) and below (green), from the 50th percentile (yellow) of the mean damages (**Panel a**) or level of intensity (**Panel b**). ** The Severity Index is calculated as the intensity of a flood by the sum of the damages (sum of the dichotomic variables affected), divided by the number of floods (for each intensity level and for the total). So that (Eq. 1):

**Severity Index** = $(\sum(\textbf{\textit{Intensity Level}} \times \textbf{\textit{Damages}})) \div \textbf{\textit{Intensity Level}}_N$.

## 4.2 Spatial Variability of floods

A detailed view on a municipal level reveals the existence of "hot spots" in number and intensity of floods (See Fig. 3). Areas of high average intensity are found along most of the coast of Andalusia and, occasionally, in some sectors of the provinces of Gerona and Tarragona. In fact, of the 20 municipalities with the highest average intensity, 12 are in Andalusia and 7 in

Catalonia. Regarding the areas with the highest number of floods, it is necessary to differentiate between two types of areas: i) large urban conurbations (Barcelona, Valencia, Malaga and Alicante) and ii) coastal spaces highly specialized in tourism (north of the province of Tarragona, province of Castellón, south of the province of Valencia and north of the province of Alicante). However, the most outstanding aspect is an opposing latitudinal gradient, since whilst the average intensity of the floods increases as we go further south, their number increases in the opposite direction. This relationship is magnified as from

the central sector of the provinces of Almería and Alicante respectively. Moreover, considering the combination of intensity and frequency, the metropolitan area of Malaga stands out as the most threatened area.



| CC.AA | Province | Pop. (2016) | Area (Km$^2$) | Density (Pop/Km$^2$) |
|---|---|---|---|---|
| **Catalonia** | *Gerona* | 243,098 | 666.2 | 365 |
| | *Barcelona* | 2,573,798 | 477.7 | 5,388 |
| | *Tarragona* | 406,536 | 1,014.4 | 401 |
| **Valencian Community** | *Castellón* | 365,002 | 920.2 | 397 |
| | *Valencia* | 1,152,087 | 710.0 | 1,623 |
| | *Alicante* | 1,085,013 | 1,663.2 | 652 |
| **Region of Murcia** | *Murcia* | 463,260 | 2,948.0 | 157 |
| **Andalusia** | *Almería* | 501,739 | 2,145.7 | 234 |
| | *Granada* | 115,177 | 447.8 | 257 |
| | *Málaga* | 1,252,694 | 1,384.2 | 905 |
| | *Cádiz* | 254,886 | 1,003.4 | 254 |
| **Total:** | | **8,413,290** | **13,380.7** | **1,200** |

**Figure 3:** Intensity average and total floods by municipality and Spanish Mediterranean Coastal Municipalities spatial summary.
The **map** shows in different colours the average intensity of the floods in each municipality and the black bars represent the total number of floods in each municipality.
The **table** reports the total population, the total area and the population density by province. The different colours represent how the values deviate above (from yellow colours to red colours) or below (from yellow colours to green colours) from the 50th percentile within the mean values of the variables (Pop., Area and Density). Source: INE, 2018.

If we analyse the variability of the data aggregated at provincial level, we see the confirmation of some of the spatial patterns detected (See Table 3). The average area affected by each flood is directly related to the differential size of the municipalities in each province. In this respect, it is appropriate to point out that in the province of Murcia the average size of the municipalities is larger, which is reflected in the fact that it is also the province where the floods have a greater spatial impact (each flood in Murcia affects an average of 574 km$^2$, compared to an average of 119 for the whole study area). In any case,





latitudinally, as from the province of Alicante, a change takes place towards floods with a larger affected area, which may also be due to climatic factors, governance or the average size of the municipalities. On the other hand, the quantity of floods that occur in each province bears a direct relationship with the size of the population exposed and its density (See Table in Fig. 3). This latter detail is especially important in the highly-developed provinces (Barcelona, Valencia and Alicante, respectively).

These provinces, together with Castellón, are those that support a higher number of floods per kilometre of coastline and they confirm the latitudinal gradient detected. This uneven N-S distribution is noticed in the intensity of the floods, i.e., the further south we go, the higher the proportion of floods of Level 2 and 3, compared to those of Level 1. Likewise, this appears to be evidenced if we consider the severity index, the expression of which is even clearer. Therefore, we can affirm the existence of a spatial pattern both in intensity and in damages in a southerly direction. In this regard, between the severity index and the

latitudinal gradient (the provinces ordered correlatively from north to south) there is a Pearson correlation of 0.91 with a significance level of 95%. Furthermore, the correlation between the percentage of L1 floods and the latitudinal gradient is -0.81. For the L2 floods the correlation is 0.73. In both cases with a significance level of 95%. This tendency may be related with the adoption of more efficient flood control measures in the northern provinces (the Catalonian and Valencian provinces) owing to their early tourism and economic development. Likewise, an explanation can also be found in the climatic factor, to

the extent that the rains are more torrential in the southern provinces (Martín-Vide, 2004). However, the clear differences between provinces of the same autonomous community invites us to consider the economic and institutional factor, since other studies have detected a growing institutional vulnerability according to the aforementioned latitudinal gradient (López-Martínez et al., 2017).





**Table 3:** Spatial flood patterns in SMCM.

| CC.AA | Province | N-Floods | N%-Floods | % Province | Severity Index | Area (Km²) | Floods/ Coast (Km²) |
|---|---|---|---|---|---|---|---|
| **Level 1** | | | | | | | |
| **Catalonia** | *Gerona* | 165 | 6.35 | 70.51 | 1.32 | 30.60 | 0.63 |
| | *Barcelona* | 520 | 20.01 | 77.15 | 1.37 | 33.41 | 3.23 |
| | *Tarragona* | 257 | 9.89 | 79.81 | 1.36 | 49.31 | 0.92 |
| **Valencian Community** | *Castellón* | 292 | 11.24 | 76.04 | 1.45 | 63.02 | 2.10 |
| | *Valencia* | 426 | 16.39 | 74.74 | 1.53 | 61.54 | 3.16 |
| | *Alicante* | 348 | 13.39 | 71.46 | 1.62 | 120.56 | 1.43 |
| **Region Murcia** | *Murcia* | 203 | 7.81 | 73.82 | 1.59 | 592.32 | 0.74 |
| **Andalusia** | *Almería* | 117 | 4.50 | 71.34 | 1.41 | 150.34 | 0.47 |
| | *Granada* | 43 | 1.65 | 58.90 | 1.26 | 70.19 | 0.53 |
| | *Málaga* | 167 | 6.43 | 56.04 | 1.41 | 158.76 | 0.80 |
| | *Cádiz* | 61 | 2.35 | 48.03 | 1.26 | 198.21 | 0.74 |
| **Total L1** | | **2,599** | **100.00** | **68.90** | **1.45** | **116** | **1.23** |
| **Level 2** | | | | | | | |
| **Catalonia** | *Gerona* | 63 | 7.10 | 26.92 | 5.37 | 30.97 | 0,24 |
| | *Barcelona* | 124 | 13.98 | 18.40 | 5.66 | 41.06 | 0,77 |
| | *Tarragona* | 55 | 6.20 | 17.08 | 6.15 | 41.56 | 0,20 |
| **Valencian Community** | *Castellón* | 88 | 9.92 | 22.92 | 5.93 | 64.21 | 0,63 |
| | *Valencia* | 139 | 15.67 | 24.39 | 7.15 | 65.84 | 1,03 |
| | *Alicante* | 123 | 13.87 | 25.26 | 6.08 | 129.33 | 0,50 |
| **Region Murcia** | *Murcia* | 59 | 6.65 | 21.45 | 6.58 | 471.98 | 0,22 |
| **Andalusia** | *Almería* | 38 | 4.28 | 23.17 | 6.37 | 169.91 | 0,15 |
| | *Granada* | 21 | 2.37 | 28.77 | 6.38 | 77.25 | 0,26 |
| | *Málaga* | 117 | 13.19 | 39.26 | 5.76 | 163.78 | 0,56 |
| | *Cádiz* | 60 | 6.76 | 47.24 | 6.77 | 159.67 | 0,73 |
| **Total L2** | | **887** | **100.00** | **26.81** | **6.18** | **118** | **0.42** |
| **Level 3** | | | | | | | |
| **Catalonia** | *Gerona* | 6 | 4.92 | 2.56 | 11.50 | 29.54 | 0,02 |
| | *Barcelona* | 30 | 24.59 | 4.45 | 11.50 | 29.16 | 0,19 |
| | *Tarragona* | 10 | 8.20 | 3.11 | 12.90 | 47.54 | 0,04 |
| **Valencian Community** | *Castellón* | 4 | 3.28 | 1.04 | 12.00 | 79.57 | 0,03 |
| | *Valencia* | 5 | 4.10 | 0.88 | 14.40 | 91.19 | 0,04 |
| | *Alicante* | 16 | 13.11 | 3.29 | 15.75 | 161.43 | 0,07 |
| **Region Murcia** | *Murcia* | 13 | 10.66 | 4.73 | 17.54 | 758.69 | 0,05 |
| **Andalusia** | *Almería* | 9 | 7.38 | 5.49 | 15.67 | 229.35 | 0,04 |
| | *Granada* | 9 | 7.38 | 12.33 | 12.67 | 53.41 | 0,11 |
| | *Málaga* | 14 | 11.48 | 4.70 | 10.29 | 260.10 | 0,07 |
| | *Cádiz* | 6 | 4.92 | 4.72 | 14.00 | 201.21 | 0,07 |
| **Total L3** | | **122** | **100.00** | **4.30** | **13.33** | **181** | **0.06** |
| **TOTAL Floods** | | | | | | | |
| **Catalonia** | *Gerona* | 234 | 6.49 | 100.00 | 2.31 | 30.67 | 0,90 |
| | *Barcelona* | 674 | 18.68 | 100.00 | 2.22 | 34.63 | 4,19 |
| | *Tarragona* | 322 | 8.92 | 100.00 | 2.15 | 47.93 | 1,16 |
| **Valencian Community** | *Castellón* | 384 | 10.64 | 100.00 | 2.28 | 63.46 | 2,76 |
| | *Valencia* | 570 | 15.80 | 100.00 | 2.59 | 62.85 | 4,22 |
| | *Alicante* | 487 | 13.50 | 100.00 | 2.77 | 124.12 | 2,00 |
| **Region Murcia** | *Murcia* | 275 | 7.62 | 100.00 | 2.82 | 574.36 | 1,00 |
| **Andalusia** | *Almería* | 164 | 4.55 | 100.00 | 2.72 | 159.21 | 0,66 |
| | *Granada* | 73 | 2.02 | 100.00 | 3.34 | 70.15 | 0,90 |
| | *Málaga* | 298 | 8.26 | 100.00 | 3.09 | 165.49 | 1,43 |
| | *Cádiz* | 127 | 3.52 | 100.00 | 3.80 | 180.15 | 1,55 |
| **Total Floods:** | | **3,608** | **100.00** | **100.00** | **2.57** | **119** | **1.71** |

*N-Floods* informs of the number of floods in each province between 1960 and 2015. *N%-Floods* reflects the percentage of the total floods that corresponds to each province. *% province* informs of the percentage of the total floods in each province that corresponds to each intensity level. *Severity Index* reflects this value for each province and for each intensity level. *Area (Km²)* and *Floods/Coast (Km²)* show for each province the km² affected and the floods which on average affect each kilometre of coast, respectively. *The different colours represent how the values deviate, above (red) or below (green), from the average (yellow) of each variable (intensity level and total).



### 4.3 Seasonal flood variability over the SMCM

The floods that affect the SMCM have considerable seasonal variability with regard to number and intensity. As is to be expected, according to the climatic conditions (Barredo, 2007; Barrera-Escoda and Llasat, 2015), the majority of the floods (58%) take place during the autumn months, especially in October owing to the recurrent "Cold Drops" that affect the region.

Furthermore, during this season there is a higher concentration as the intensity increases (55% of Level 1, 65% of Level 2 and 74% of Level 3). Therefore, autumn (September-November) is the season with the most danger of flooding, with regard to both quantity and intensity. Autumn is followed by winter (December-February), summer (June-August) and finally spring (March-May). (Fig. 4 Panel a). However, there is little difference between the intensity of the autumn and winter floods (Fig. 4 Panel b) (mean intensity of 1.34 in the case of winter compared to 1.36 during the autumn). In spring, on the other hand, and

especially summer, the average intensities are lower (mean intensity of 1.22 and 1.16 respectively). The high intensity of the winter floods is probably related to the type of atmospheric situation that generates these floods and which usually leads to large accumulations of rainfall over several days (Muñoz-Diaz and Rodrigo, 2004). Related to this point, the climatic patterns will be studied in depth in successive studies within this same project.

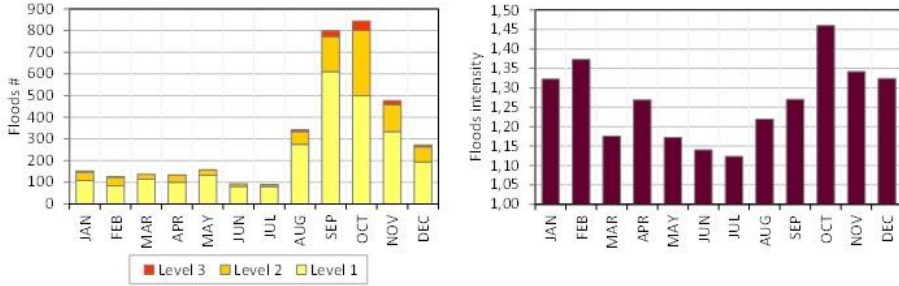

**Figure 4:** Monthly distribution of flood frequency and average intensity in SMCM.

This seasonal variability presents notable differences between provinces (See Fig. 5). Regarding the number of floods, whilst the autumn seasonal pattern mentioned above is reinforced for the provinces of the east coast, in the southern provinces autumn loses importance in favour of winter, which is the season that concentrates a higher number of floods. This reveals a rainfall pattern associated with intense rains owing to the variability of the polar front (Muñoz-Diaz and Rodrigo, 2004) which

especially affects the provinces of the south-west of the study area. However, these latter are also not exempt from the impact of the frequent synoptic situations associated with an easterly flow that affect the rest.

A similar spatial distribution can be observed in the average monthly intensity per province. In the provinces of Granada and Malaga the autumn floods present the highest intensity values of the study area. However, unlike the frequency, the distribution pattern seems less clear. There are coincidences in that the autumn and winter months are those that record floods of a more

intensive nature, mainly in the provinces between Castellón and Granada, inclusive. However, in the provinces of Gerona and Cadiz, the mean intensity is higher in the winter months. Finally, in the provinces of Barcelona and Tarragona, late summer floods also have a significant importance in terms of flood frequency. This situation is consistent with the results of previous works in these provinces (Llasat et al., 2013a, Gilabert and Llasat, 2018).



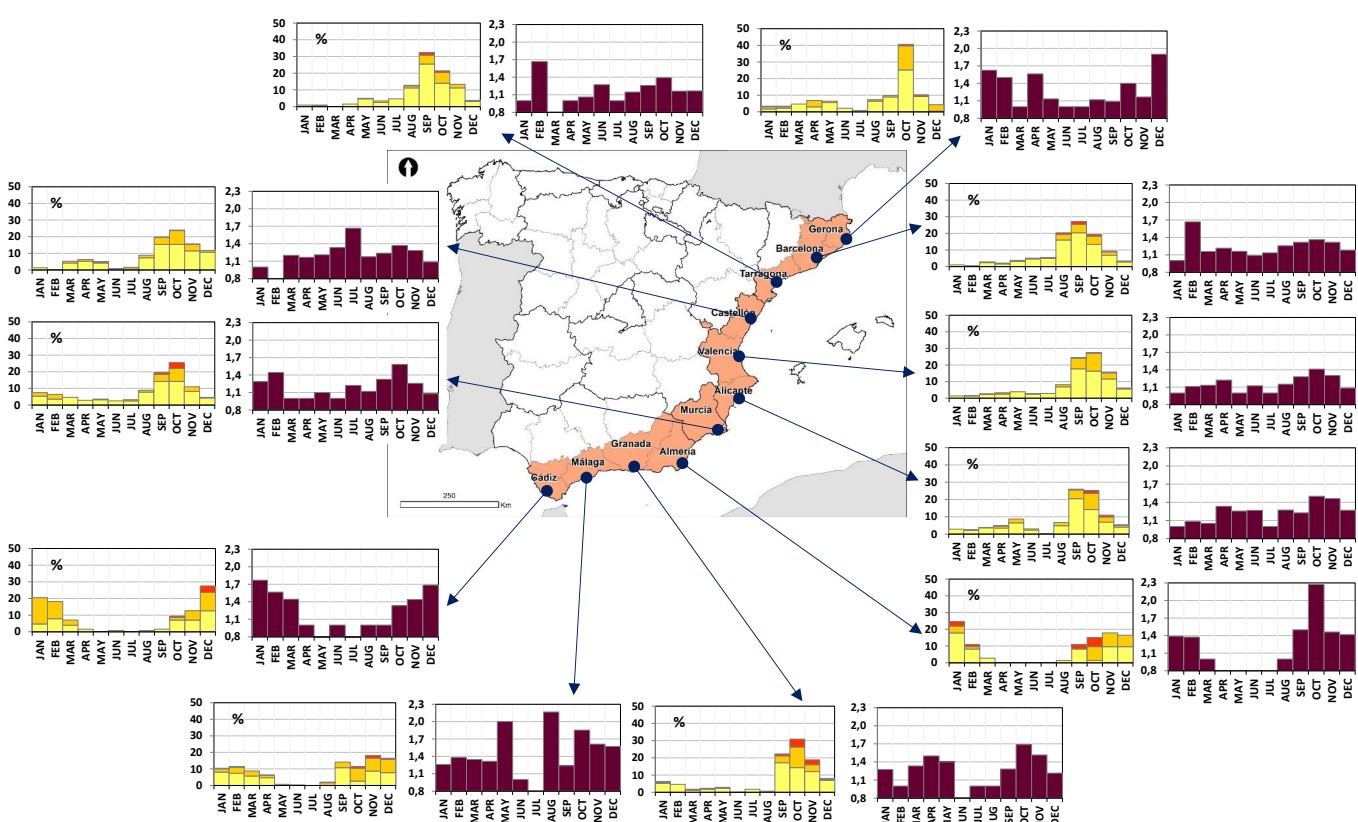

**Figure 5:** Spatial variability of flood frequency and average intensity in SMCM by province.





### 4.4 Flood damage variability

The amount of damages reported and their type also present a notable spatial variability. With regard to the total amount of damages per province (Fig. 6 Panel a) the size of the exposed population is the main factor to explain it. In fact, the provinces with a higher population (Barcelona, Valencia and Alicante) are those that report a higher number of total damages (Fig. 6

Panel a). Furthermore, if we compare the total number of floods per province with the total damages reported (average quantity of damages reported per flood) the latitudinal gradient referred to above continues to be reflected. That is, whereas in the provinces to the north of Valencia the average type of damage reported per flood is lower than 2, from Valencia towards the south this value is higher.

With regard to the different types of damage per province (Fig. 6 Panel b) the highest quantity of damages affects roads and
buildings, the sum of which represents over 60% of all the damages reported. The provinces of Barcelona, Alicante and Malaga are highly developed and specialized in activities in the service sector and tourism, owing to which the damages related to the primary sector are low. However, in the provinces of Tarragona, Almería and Granada, where the agricultural sector continues to have an important comparative economic weight, the damages in this sector are considerable.

Lastly, the latitudinal gradient is also observed in the analysis of the damages. This time it is explained in the record of injured
persons and fatalities. As we head further south, there is an alarming increase in the percentages of these variables from 3% on average between Gerona and Alicante to 6% between Murcia and Andalusia.

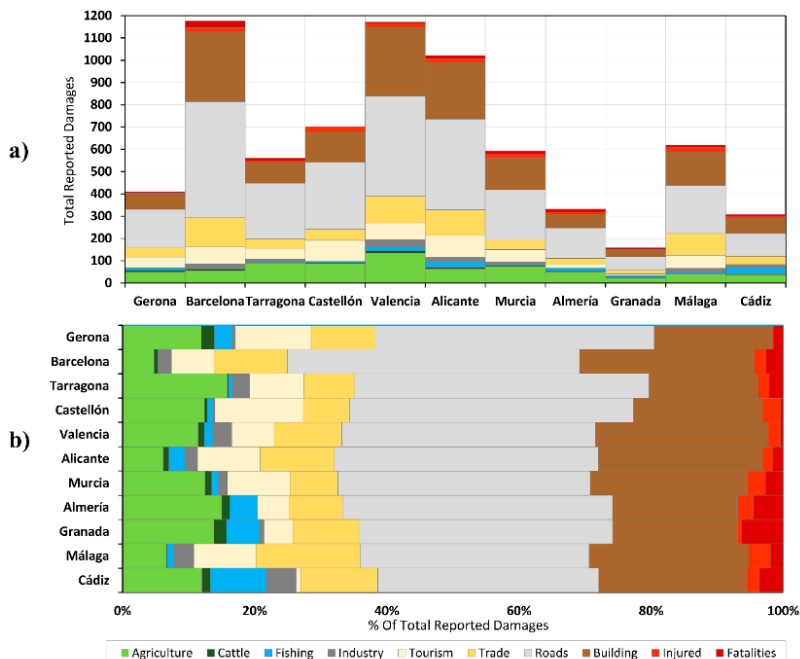

**Figure 6:** Types of damages by province.
**Panel a** shows the total damages per type and province. **Panel b** gives the damages per type, as a percentage of the total damages in each
province.



## 4.5 Flood evolution and trends

The floods in the SMCM present an annual variability in frequency, impact and intensity closely related with the variation in the frequency and intensity of precipitations (Martin-Vide, 2004).

With regard to the annual mean intensity (See Fig. 7, Panel a) from 1960 to 1994 the data were more variable and extreme. In fact, the 10 years with greatest average intensity took place during that period. Furthermore, the five years of highest mean intensity were, respectively: 1973, 1987, 1962, 1965 and 1961. However, it is important to emphasize that the mean annual intensity presents a statistically significant negative trend, i.e., the average intensity of the floods has descended during the period analysed. Likewise, despite the Severity Index during the first five years of the nineteen-sixties was particularly high, their average annual value also presents a significant negative trend. According to the Severity Index rank, the following years should be highlighted: 1973, 1962, 1987, 1982 and 1964. Some of the highlighted years coincide with the most catastrophic floods occurring in the study area (1973, 1962 and 1987) in the last century.

With regard to the annual frequency of the floods (See Fig. 7, Panel b) an increase is observed since the eighties and, especially, since 1996. Since then, in the majority of years the number of floods is above the average. The trend analysis detects a statistically significant positive trend, which reveals that every year the floods increase by 2.3% compared to their average value. However, this increase is not homogenous according to the intensity level, since whilst the Level 1 and Level 2 floods present significant positive trends, those of Level 3 remain stable, or with no appreciable trend. The rising trend is more pronounced in the case of the Level 1 floods, with an annual increase of 2.8%, compared to 1.1% in the case of Level 2 floods. Turco and Llasat (2011) and Llasat et al., (2010), have also find out a floods trend of recent decades in Catalonia that is due mainly to the increase in urbanization in flood-prone areas near torrential and non-permanent streams.

Lastly, the evolution of the area affected presents a similar variability to the behaviour of the frequency of the floods (See Fig. 7, Panel c). There is a coincidence in pointing to the decade of the nineteen-eighties as the time when the values began to increase. Especially outstanding is the second half of the nineteen-eighties and the decade of 2000, as especially disastrous periods. However, it is necessary to point out some nuances and differentiations. The area affected has a positive trend for the total values, but if we consider the size of the area affected according to intensity levels, a significant positive trend has only been detected for Level 1 floods. That is, every year the area affected by the floods increases by 166 km$^2$ (an increase of 2.2%). This situation takes on special importance in the case of Level 1 floods, for which the new area affected is 158 km$^2$ per year (a growth rate of 2.9%).





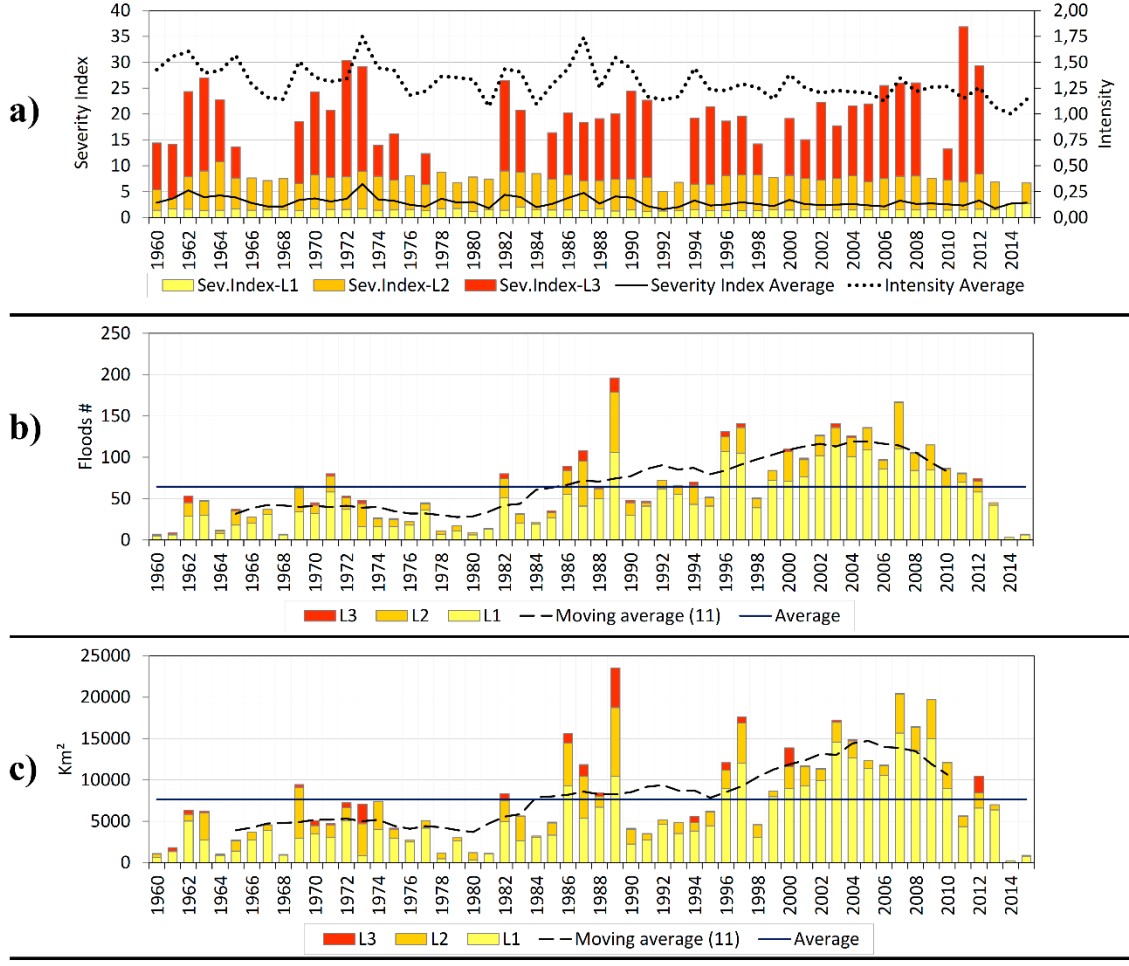

**Figure 7:** Temporal variability of the intensity, severity and spatial impact of floods in the SMCM.
The values inform of the annual accumulation per intensity level. The mobile mean of 11 years has been added to temper the variability, as well as the total mean of each variable, in order to identify years with values above or below the mean.

**Panel a** informs on the variability of the annual intensity and severity of the floods. **Panel b** informs on the variability of the annual number of floods. Finally, **Panel c** informs on the variability of the area affected.

Related to the positive trend observed in the data, Llasat et al. (2016) pointed out it may be due to main factors: i) climatic issues (a greater recurrence of torrential rain events in the study area), and ii) the increase in exposure and vulnerability due to the increase of population and economic growth.

The fact that the floods of L1 consider not only river floods (also consider flash floods and in situ floods), can magnify the importance of the increase in exposure, as to the growth of the exposed surface in flood zones. Thus it is necessary add the growth of the surface outside the floodplain (Pérez-Morales et al., 2018). The evidence about the last is that L1 floods are the ones with the most clear upward trends.

However, there are other factors that should not be overlooked or dismissed. According to Llasat et al., 2009, it is important

to consider that trends may be biased: i) a greater sensitivity or perception towards natural risks from the public opinion could



increses news about floods in newspapers; ii) a greater spatial coverage of the news thanks to the improvement of communications. The complexity of the factors (social, cultural, environmental and perceptual) involved in flood processes make us to think about the influence of the mentioned items in the observed trends. Therefore, a deeper knowledge on climatic, geographic and socioeconomic variables involved is necessary. This will be the objective of successive research projects.

## 5 Conclusions

The correct knowledge of the real situation of the territories in the light of flooding is of capital importance with a view to presenting optimum resilience and future adaptation. In this respect, high-resolution spatiotemporal flood databases are a key tool which, in the hands of spatial managers and scientists, may contribute to improving the situation in especially sensitive regions. In this respect, the municipalities of the Mediterranean Coast of mainland Spain are an area especially vulnerable to floods.

From the findings, we can draw some important conclusions. The SMC-Floods database represents the flood database with the highest spatiotemporal resolution developed for the study area. The results have enabled the reconstruction of 3,008 cases of flooding that affected all the municipalities studied during the last 55 years.

The exploitation of the database has made it possible to obtain a series of values that provide evidence of trends which reveal the socio-environmental dynamic. In this respect, the damages show that the major impacts occur in roads, buildings, agriculture and trade. Furthermore, the average area affected per flood is 119 km$^2$. In general, the months that pose the greatest hazard with regard to the number of floods and their intensity are the autumn months, although the winter is also a highly hazardous season.

The detailed spatial analysis has enabled us to identify a series of black spots where the intensity of the floods and the amount of damage are very high (especially on most of the coast of Andalusia and in some areas of Gerona and Tarragona). Furthermore, there are places where the large size of the population exposed determines a high recurrence of the floods.

However, the main conclusion observed in general in the majority of the variables and indices considered is the presence of a clear latitudinal gradient, characterized by more severe, intensive, extensive and damaging floods as we move from north to south. This spatial inequality is foreseeably explained by greater deficiencies in the spatial planning of the provinces in the south, although the climatic and orographic factors cannot be ruled out. Under these circumstances the southern municipalities of the Mediterranean coast of the Iberian Peninsula are the places where the biggest and best adaptation plans should be implemented, especially if it is taken into account that in these provinces there is also a greater risk of mortality associated with floods. It is understood, therefore, that without correct planning and adaptation, these places face a critical situation where the relationship between human and natural systems is called into question and compromises the future development and safety of the inhabitants.

Lastly, it is important to highlight that the intensity and mean annual severity of floods have undergone a statistically significant negative trend. That is, on average floods tend towards a lower intensity and severity. However, the annual frequency and the average area affected by the floods has experienced a positive trend. Nonetheless, this increase is not homogenous according




to intensity level, since whilst the Level 1 and Level 2 floods present significant positive trends, those of Level 3 remain stable. In this respect, a paradox is revealed, insofar as it is positive that the more catastrophic floods do not increase, however with the current flood management tools (structural and non-structural) it would be feasible for the trends detected to be negative. That is, in the light of a negative panorama, we consider the lesser of the ills to be positive.

*Acknowledgements.* We thank the LV and ABC newspapers, for their open data access policy. Special thanks to the LVM and EMV newspapers for collaborating on this research project by allowing free access to their databases. This work has been partially supported by the Spanish Ministry of Economy and Innovation (CGL2016-75996-R). Thanks also to Interuniversity Institute of Geography for the financial support. SG-G acknowledges the support of the Spanish Ministry of Science,
Innovation and Universities through "Juan de la Cierva-Incorporación" grant (IJCI-2016-29016).

*Competing interests.* The authors declare that they have no conflict of interest.

*Data availability.* The systematic data of SMC-Floods database are not publicly accessible because they are currently being
used in an ongoing research project. Aggregate data at provincial level can be obtained by request addressed to the corresponding author.

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
