# Peer review of "SMC-Flood database: A high resolution press database on flood cases for the Spanish Mediterranean Coast (1960-2015)"

_Natural Hazards and Earth System Sciences, 2019_

## Referee Comment (RC1) · Anonymous Referee #1 · 3 Mar 2019

General comments This is a very interesting paper, well-structured and written. The issues addressed are within the scope of NHESS. The construction of the SMC db is of high importance for the analysis of the spatial and temporal changes in the coastal areas vulnerability to floods. Conclusions can be very useful to decision-makers for adaptation planning. The methodology followed for the development of the SMC database is appropriate and well presented. Therefore, the article merits being published, with minor changes. Specific comments 1. My only scientific concern is the use of 2 different averaged indices for the average impact severity. I mean, the intensity level is actually related to the impact magnitude: low damages / major / deaths and/or general destruction. Then the authors produce a damage severity index, which uses in the equation

the intensity level as weight implemented on the various damages occurrence. I can understand that cases can be compared better based on the severity index. However, I am not convinced about the use of 2 'average' values used to evaluate trends or vulnerability at aggregated areas. What is the point? Maybe this could be better explained. 2. In what concerns the structure of the paper, my only concern is the introduction. In page 4, paragraph 4 (lines 24-29) is too methodological to be included in the introduction. It confuses the reader who expects to read the objectives and research questions instead of fragmentary information about methods employed. I suggest this part to be transferred to the methods section. 3. Please consider for your references regarding the databases in other countries also the high-impact weather events database of the National Observatory of Athens, Greece, which is also active on-line, constantly updated and with weather and impact intensity classification (10.5194/nhess-13-727-2013). The NOA db has been also based on press articles. Technical corrections 1. P3, l15: Please delete the 'y'. 2. P4, l10: Please delete 'but'. 3. P4, l14: It is 'flood cases', not 'floods cases'. Please repeat correction throughout the article. 4. P4, l15: Please correct as: In this regard it must be clarified the difference between flood cases and flood events 5. P4, l24: Please add 'digital' before 'archives'. 6. P4,l29: The sentence is too big. Please start a new one from 'in general...' 7. P4, l33: Please delete 'This is', otherwise the sentence does not make sense. In the same sentence, please use the same term throughout the paper regarding the flood 'case'. You have explained very well in the document the difference between case and event. So, the words 'episodes' and 'events' in this sentence do not fit. 8. P5, l17: Please correct as 'emphasized'. Also, please rephrase the entire sentence as it is not clear, especially the second part. 9. P6, l4: Please explain the: (2003: 800) 10. P6, l5: please correct as: These situations 11. P6, l10: please use the same term: environmental or climatic 12. P6, l13: just a thought: is this sentence for Franco necessary? 13. P6, l18: please cut this sentence in 2 parts. it is too big and difficult to read 14. P6, l19: it is weird the use of 'it has...' after 'the following criteria'. I think it can be improved. 15. Table 1: what is MEDIFLOOD? do you mean SMC-Flood db? Also, the authors could enter an

extra column to report the cities of head offices. The full newspaper names could be added here as a comment. 16. P6, last paragraph: the different names are confusing. Consider keeping the short names of Table 1 everywhere in the text. 17. P7, l5: 1) the sentence is too big. Please enter full-stop before 'Taking into account'. 2) Please consider avoiding the footnote since it concerns only one source. You could include it in the text instead. 18. P7, l7:correct as 'validated' 19. P7,l8: please delete 'de'. L10: please add space before 'Secondly'. L22: please correct as 'within' 20. P10, l4: Please begin a new sentence at 'Multiplying...' 21. Table 2: maybe it is better if you write 'average severity index' 22. P11, l7: Please add 'the' before 'number' 23. P11, l8: please consider defining 'average intensity', as this is the first time we read this. 24. P15, l5: 'concentration' of what? I think something is missing 25. P19, l11: Please add 'to' before 'add' 26. P20, l3: please correct as 'makes us'

---

## Referee Comment (RC2) · Anonymous Referee #2 · 5 Mar 2019

General comments: The paper "SMC-Floods database: A high resolution press database on floods for the Spanish Mediterranean Coast (1960-2015)" provides a preliminary description and analysis of flood data collected from press news. It is not a novel initiative at European or Spanish level, but it comprises a large extension of a flood damage prone region. I admit that such effort merits publication somewhere, but not in the present format, which requires a major review before it can be published. In scientific outcomes are highly bias by the journalist judgment of the flood damages and newspaper coverage and audience. Therefore, caution should be placed on the interpretation of the data. There is not a critical analysis of the results in relation to other more robust database, for instance the analysis from the National Insurance Consortium. As this database reflects risks (mostly exposure and vulnerability), most of the hydroclimatologic trends and changes on hydroclimatic conditions may not be valid.

There are a number of points that the authors should correct. 1.- The manuscript requires a detail English correction on the style. It looks a direct translation from a Spanish text, I would say that the authors used google translator, otherwise, I cannot explain the use of some very incorrect terms. Among the most critical one are "Cold Drop" cited in the paper, and probably authors refer to "cold pool" or "mean mobile" (cited in figure 7) instead of "moving average". These are only few examples, but the text is full of informal terms or sentences that do not make any sense in English. 2.- The manuscript is very long and this makes difficult to read. The authors should analyze in each sentence and use proper language addressing the point in a direct way. 3.- Several sections can be shortened, including the introduction and conclusions. 4.- Abstract: The abstract should be completely re-written. The way it is written looks and introduction rather than a summary. Sentences such as "Floods are the natural disaster that affects the greatest number of people and causes the highest economic losses in the world" are fine for the introduction, but not for the abstract. Please, start the abstract by telling the reader at once what the paper is: new data, a review of progress, a new technique, a synthesis, or whatever describes the nature of the paper. Unnecessary descriptive phrases and qualifiers should be left out of the abstract. Write the abstract as styled summary of its essential information; and include as much specific information as possible on the results. 5.- Introduction: There is a long description of flood databases from press news in Europe and the world, and they do not provide any key information to objective or analysis to be addressed by the MSC database. I would suggest leaving only the most relevant databases, and includes the rest on a table indicating the country, region, time period covered, data source, type of data included, authors. 6.- Page 4. Introduction Lines 24 to 30 I suggest to move to methodology section 7.- Page 4 introduction. Lines 31 to end of section, I suggest to delete this paragraph. Instead you should describe the specific objectives of this study. 8.- Page 6, lines 13 to 15 probably not needed, delete. 9.- Page 8. Indicate the

list of damage types in a single line. 10.- Type of damages. Here roads and housing are the most common ones. I wonder if the news are bias to these two types because of most easy ones to be reported right after the event. 11.- page 10 line 25. How the quantity of damage was calculated?. In the case of housing, are you reporting the number of affected houses, or on roads, the number of cut roads. . .? 12 Page 15. "cold drops" is a direct translation of the Spanish informal term. Please, use "cold pool" or mesoscale convective sytems. 13.- Page 15. From line 20 to 28, it is poorly written and they need major changes. 14. Page 17. Line 6. I don't understand "the latitudinal gradient referred to above continues to be reflected. 15. Page 18. Line 25-27. This is not suprising due to the press nature of the database. As more small villages are cited on the newspaper, the flood extend on the database increases. 16 Page 19. Lines 10-11. The sentence "The fact that the floods of L1 consider not only river floods (also consider flash floods and in situ floods), can magnify the importance of the increase in exposure, as to the growth of the exposed surface in flood zones." I wonder if the main problem is the nature of the database, because social perception of risk increase with time, since any single damage is reported on the local news. 17.-Conclusions should go to the point of the main results. In the present format, there are too long, and they should be shortened.

In summary, the manuscript requires major changes before the manuscript can be published. The authors should made major improvements on the English edition, including major rewritten of the manuscript. Several sections should be reduced according to the above changes.

Other minor changes are suggested on the pdf document.

Please also note the supplement to this comment:
https://www.nat-hazards-earth-syst-sci-discuss.net/nhess-2019-10/nhess-2019-10-RC2-supplement.pdf

[Figure]

**Supplement:**

[revised manuscript text omitted]

---

## Author Comment (AC1) · 20 May 2019

**Referee #1 responses:**

**Overall comment:**

- **REVIEWER GENERAL COMMENT:** This is a very interesting paper, well-structured and written. The issues addressed are within the scope of NHESS. The construction of the SMC db is of high importance for the analysis of the spatial and temporal changes in the coastal areas vulnerability to floods. Conclusions can be very useful to decision-makers for adaptation planning. The methodology followed for the development of the SMC database is appropriate and well presented. Therefore, the article merits being published, with minor changes.
- **AUTHORS RESPONSE:** Thank you for your flattering and constructive comments. We believe that the results of this work may represent an improvement in the knowledge of the spatio-temporal patterns of floods in the Spanish Mediterranean coast in the last 50 years. We have included all your suggestions in the new manuscript version, this will make the manuscript undoubtedly more robust.

**Specific comments:**

- **REVIEWER COMMENT:** 1. My only scientific concern is the use of 2 different averaged indices for the average impact severity. I mean, the intensity level is actually related to the impact magnitude: low damages / major / deaths and/or general destruction. Then the authors produce a damage severity index, which uses in the equation the intensity level as weight implemented on the various damages occurrence. I can understand that cases can be compared better based on the severity index. However, I am not convinced about the use of 2 'average' values used to evaluate trends or vulnerability at aggregated areas. What is the point? Maybe this could be better explained.
- **AUTHORS RESPONSE:** We agree with you that the use of two different means can lead to confusion. However, we believe that the comparison between these two indices can help to have a better idea of the problems generated by floods in the study area. We consider that the average intensity is a robust measure, since it limits the accumulated bias of adding several quantitative magnitudes extracted from qualitative information. In addition, the average intensity is based on numerous works that use the three levels of intensity for the study of floods through historical and hemerographic documentation (Camuffo and Enzi, 1996; Barriendos et al., 2003; Llasat, et al., 2005; Barriendos et al., 2014). However, we believe that the severity index provides additional information, although it may be subject to greater subjectivity. As an example, a flood of intensity 2 could produce major damage, but with concentrated effects in agriculture (Intensity = 2, severity index = 2), while a flood of intensity 1, could have some weak damages, but extended to a large number of sectors (for example roads, tourism, commerce and agriculture) (Intensity = 1; severity index = 4 * 1 = 4). In other cases a flood could be very intense and also affect a large number of sectors, so its final impact is greater than if simply considering the intensity (For example, a flood of intensity 2, which affected roads, agriculture, tourism and trade would have a severity index of 8). Therefore, the severity index offers information that is complementary to the intensity level and the amount of damages.

- **REVIEWER COMMENT:** 2. In what concerns the structure of the paper, my only concern is the introduction. In page 4, paragraph 4 (lines 24-29) is too methodological to be included in the introduction. It confuses the reader who expects to read the objectives and research questions instead of fragmentary information about methods employed. I suggest this part to be transferred to the methods section.
- **AUTHORS RESPONSE:** Following your recommendation, we have transferred this part to the methodology section of the manuscript.

- **REVIEWER COMMENT:** 3. Please consider for your references regarding the databases in other countries also the high-impact weather events database of the National Observatory of Athens, Greece, which is also active on-line, constantly updated and with weather and impact intensity classification (10.5194/nhess-13-727-2013). The NOA db has been also based on press articles.
- **AUTHORS RESPONSE:** Thank you very much for your important suggestions. We have included the reference 10.5194 / nhess-13-727-2013 in the introduction section and we have pointed out the peculiarities of this database. Additionally, we have added its hemerographic characterization to the description of the NOA database.

**Technical corrections:**

- **AUTHORS RESPONSE:** Thank you very much for highlighting these important details and providing advice about the convenience of including some necessary clarifications. All these issues have been taken into account. Most of the corrections have been resolved thanks to a native English speaker who will be responsible for reviewing the text of the new version before being sent.

Here is a more detailed description of the changes carried out:

- **REVIEWER COMMENTS:**
    1. P3, l15: Please delete the 'y'.
        - **AUTHORS RESPONSE:** Delete

    2. P4, l10: Please delete 'but'.
        - **AUTHORS RESPONSE:** Delete

    3. P4, l14: It is 'flood cases', not 'floods cases'. Please repeat correction throughout the article.
        - **AUTHORS RESPONSE:** this mistake has been corrected

    4. P4, l15: Please correct as: In this regard it must be clarified the difference between flood cases and flood events
        - **AUTHORS RESPONSE:** this mistake has been corrected

    5. P4, l24: Please add 'digital' before 'archives'.
        - **AUTHORS RESPONSE:** word included

    6. P4,l29: The sentence is too big. Please start a new one from 'in general.
        - **AUTHORS RESPONSE:** we have corrected this sentence in the revised manuscript.

    7. P4, l33: Please delete 'This is', otherwise the sentence does not make sense. In the same sentence, please use the same term throughout the paper regarding the flood 'case'. You have explained very well in the document the difference between case and event. So, the words 'episodes' and 'events' in this sentence do not fit.
        - **AUTHORS RESPONSE:** Thank you very much for your corrections. We have considered your suggestions and have replaced events for cases.

    8. P5, l17: Please correct as 'emphasized'. Also, please rephrase the entire sentence as it is not clear, especially the second part.
        - **AUTHORS RESPONSE:** we have corrected this sentence in the revised manuscript.

9. P6, l4: Please explain the: (2003: 800)
  - **AUTHORS RESPONSE:** we have deleted ":800".

10. P6, l5: please correct as: These situations.
  - **AUTHORS RESPONSE:** this mistake has been corrected

11. P6, l10: please use the same term: environmental or climatic
  - **AUTHORS RESPONSE:** we have replaced environmental by climate

12. P6, l13: just a thought: is this sentence for Franco necessary?
  - **AUTHORS RESPONSE:** probably this sentence is not necessary. So, we have deleted this sentence.

13. P6, l18: please cut this sentence in 2 parts. it is too big and difficult to read
  - **AUTHORS RESPONSE:** we have corrected this paragraph and shortened the sentences in the revised manuscript.

14. P6, l19: it is weird the use of 'it has:' after 'the following criteria'. I think it can be improved.
  - **AUTHORS RESPONSE:** we have corrected this sentence in the revised manuscript.

15. Table 1: what is MEDIFLOOD? do you mean SMC-Flood db? Also, the authors could enter an extra column to report the cities of head offices. The full newspaper names could be added here as a comment.
  - **AUTHORS RESPONSE:** we have corrected the mistake and replaced MEDIFLOOD by SMC-Floods database. We have also added an extra column in the table to report the head offices cities. Additionally, the full newspaper names are added as a footnote to the table.
  -

16. P6, last paragraph: the different names are confusing. Consider keeping the short names of Table 1 everywhere in the text.
  - **AUTHORS RESPONSE:** We have considered your suggestion and for the sake of clarity, we have kept the short names of Table 1 throughout the text.

17. P7, l5: 1) the sentence is too big. Please enter full-stop before 'Taking into account'. 2) Please consider avoiding the footnote since it concerns only one source. You could include it in the text instead.
  - **AUTHORS RESPONSE:** we have corrected this paragraph and shortened the sentences in the revised manuscript. Additionally, we have added the reference of note 1 in a sentence within the main text.

18. P7, l7:correct as 'validated'
  - **AUTHORS RESPONSE:** this mistake has been corrected

19. P7,l8: please delete 'de'. L10: please add space before 'Secondly'. L22: please correct as 'within'
  - **AUTHORS RESPONSE:** all this mistakes has been corrected

20. P10, l4: Please begin a new sentence at 'Multiplying:

– **AUTHORS RESPONSE:** we have corrected this paragraph and shortened the sentences in the revised manuscript.

21. Table 2: maybe it is better if you write 'average severity index'
    – **AUTHORS RESPONSE:** we have added this suggestion

22. P11, l7: Please add 'the' before 'number'
    – **AUTHORS RESPONSE:** "the" has been added

23. P11, l8: please consider defining 'average intensity', as this is the first time we read this.
    – **AUTHORS RESPONSE:** following this suggestion, the definition of average intensity has been added to this paragraph

24. P15, l5: 'concentration' of what? I think something is missing
    – **AUTHORS RESPONSE:** Thanks for detecting this mistake. In this sentence we want to show that during the fall there is a greater concentration of floods as intensity increases. Therefore we have modified the phrase to include the word floods: "during this season there is a higher concentration of floods as the intensity increases".

25. P19, l11: Please add 'to' before 'add'
    – **AUTHORS RESPONSE:** "to" has been added

26. P20, l3: please correct as 'makes us'
    – **AUTHORS RESPONSE:** this mistake has been corrected

---

## Author Comment (AC2) · 20 May 2019

**Referee #2 responses:**

**Overall comment:**

**Reviewer General Comment:**
The paper "SMC-Floods database: A high resolution press database on floods for the Spanish Mediterranean Coast (1960-2015)" provides a preliminary description and analysis of flood data collected from press news. It is not a novel initiative at European or Spanish level, but it comprises a large extension of a flood damage prone region. I admit that such effort merits publication somewhere, but not in the present format, which requires a major review before it can be published.

**Authors Response:**
Thank you for your suggestions, we have included all of them in the new manuscript version. According to your comments, different European research groups are faced with the effort to collect and organize large databases for extreme hydrometeorological events on a historical scale. In Spain, there are at least two working groups that collect events and cases of flood from historical periods until nowadays and try to cover the entire Spanish territory through different methodologies. The focus of our work is to cover two main needs:
1. Analyze the trends of flood cases and events in the Spanish Mediterranean coast. This is an area that has increased the number of floods and, according to the IPCC (2012), there are great uncertainties about the importance of the physical factor and the human factor in the balance of economic losses caused by floods.
2. The final goal of this study is focused on knowing to what extent the variability of floods is caused by changes in the social systems. In this regard, we consider that other floods databases show a lack of data of floods in small towns (Paprotny et al., 2018) and we show that the lack of information has a substantial impact on observed trends.

**Reviewer General Comment:**
In scientific outcomes are highly bias by the journalist judgment of the flood damages and newspaper coverage and audience. Therefore, caution should be placed on the interpretation of the data.

**Authors Response:**
We share your view of this point, for this reason the manuscript included a paragraph of assumption of limitations that takes into account the possible bias indicated by the reviewer: "*However, there are other factors that should not be overlooked or dismissed. According to Llasat et al., 2009, it is important to consider that trends may be biased by various reasons: i) a greater sensitivity or perception towards natural risks from the public opinion could increase news about floods in newspapers; ii) a greater spatial coverage of the news thanks to the improvement of communications. The complexity of the factors (social, cultural, environmental and perceptual) involved in flood processes make us think about the influence of the mentioned items in the observed trends. Therefore, a deeper knowledge on climatic, geographic and socioeconomic variables involved is necessary*".

To the above explanation we have added another possible biased commentary. Eisensee and Strömberg (2007) argue that the coverage of natural disasters in the press depends on the availability of other newsworthy material at the time of the disaster. Additionally, and as you point out, there may be some spatial bias in the news based on the newspaper's spatial coverage (Walmsley, 1980). In this regard, and as explained in section 3 (Methodology and sources), the newspapers used are regional newspapers, which specifically cover the information of each of the Autonomous Communities studied throughout the study period.

On the other hand, the experience of other related works (Llasat et al., 2009) and the way of gathering information, show that the subjectivity of the journalist or the newspaper can bias the level of intensity, but not the type of damage. In this regard, the type of damage may be under-documented, but it can rarely be over-documented. However, it is true that the increase of population has been able to influence the increase of news in small populations. This is precisely one of the facts that we highlight in this work, since a large part of the detected trends are influenced by the population increase. This fact can be observed when we analyze the trends in floods according to population growth. In making this analysis, we observed that the greater the population increase of the municipalities between 1960 and 2011 was (the two extreme census moments during the study period), the greater significance and intensity the trend detected has. In the following table (Table 1) it can be observed that in the set of municipalities where population have grown less than 50% between 1960 and 2011, the floods have no significant trend. However, floods have a statistically significant trend in municipalities that have grown more than 50%. The interesting thing is that the rate of increase in floods is greater as the population growth is greater. This shows that the detected trends are largely influenced by the increase in exposure.

**Table 1:** Floods trends in Spanish Mediterranean Coastal Municipalities related to ranges of population increase between 1960 and 2011.

| Population increase range in % | Kendall's tau | P-value | Sen´s Slope |
|---|---|---|---|
| Less than 0% | -0.048 | 0.733 | 0 |
| Between 0 and 50% | -0.060 | 0.550 | 0 |
| More than 50 and less than 100% | 0.280 | 0.005 | 0.127 |
| More than 100 and less than 200% | 0.340 | 0.000 | 0.286 |
| More than 200 % | 0.380 | < 0,0001 | 0.471 |

*To calculate trends, we have used Hirsch and Slack's nonparametric test (1984), which is based on Mann-Kendall range. The trial version of XLSTAT software (Addinsoft, 2018) was used to calculate it. The Mann-Kendall test provides a level of statistical significance (p-value). The threshold of significance chosen was 95%, which indicates that p-values above 0.05 should lead to rejecting the hypothesis of a trend in the series. When the p-value is less than 0.05, the trend can be positive or negative. Sen´s Slope shows the annual change rate in floods. That is, the value informs about the annual increase or decrease of the floods.*

To clarify this point, we have added the previous table and part of the previous comments to section 4.5. of the manuscript. However, it is true that there are still many uncertainties regarding the possible information bias. For this reason, in the conclusions section we have qualified that, although the results are robust when relating population increase and increase in floods, the tendencies detected may be biased by journalistic issues.

**Reviewer General Comment:**
There is not a critical analysis of the results in relation to other more robust database, for instance the analysis from the National Insurance Consortium. As this database reflects risks (mostly exposure and vulnerability), most of the hydroclimatologic trends and changes on hydroclimatic conditions may not be valid.

**Authors Response:**
We agree with the reviewer. In the new version of the work, a critical analysis will be included in relation to the National Insurance Consortium database in order to show the possible similarities and differences with our database. However, it should be noted that the Spanish insurance contract law does not require the insurance of the home. Therefore, this database could be even more biased than ours depending on the degree of insurance coverage in the municipalities of the study area (Clavero, 2016). On the other hand, the fact

that the National Insurance Consortium database is based on private insured assets limits information on the impact of floods on public goods such as roads. In this regard, and taking into account the great weight that road damages have on our database, it is not surprising that the SMC-Floods database considers a number of cases far superior to the National Insurance Consortium database.

**Reviewer Specific Comment:**
1.- The manuscript requires a detail English correction on the style. It looks a direct translation from a Spanish text, I would say that the authors used google translator, otherwise, I cannot explain the use of some very incorrect terms. Among the most critical one are "Cold Drop" cited in the paper, and probably authors refer to "cold pool" or "mean mobile" (cited in figure 7) instead of "moving average". These are only few examples, but the text is full of informal terms or sentences that do not make any sense in English.

> **Authors Response:**
> A native English speaker will be responsible for reviewing the text of the new version before being sent.

**Reviewer Specific Comment:**
2.- The manuscript is very long and this makes difficult to read. The authors should analyze in each sentence and use proper language addressing the point in a direct way.

> **Authors Response:**
> The deep revision of the language of the manuscript by a native English speaker, has been an important summary and synthesis of the manuscript. In this way, we consider that the ideas are now expressed more clearly.

**Reviewer Specific Comment:**
3.- Several sections can be shortened, including the introduction and conclusions.

> **Authors Response:**
> The same than the last point.

**Reviewer Specific Comment:**
4.- Abstract: The abstract should be completely re-written. The way it is written looks and introduction rather than a summary. Sentences such as "Floods are the natural disaster that affects the greatest number of people and causes the highest economic losses in the world" are fine for the introduction, but not for the abstract. Please, start the abstract by telling the reader at once what the paper is: new data, a review of progress, a new technique, a synthesis, or whatever describes the nature of the paper. Unnecessary descriptive phrases and qualifiers should be left out of the abstract. Write the abstract as styled summary of its essential information; and include as much specific information as possible on the results.

> **Authors Response:**
> We appreciate the comment of the reviewer and we have rewritten completely the abstract following his suggestions.

**Reviewer Specific Comment:**
5.- Introduction: There is a long description of flood databases from press news in Europe and the world, and they do not provide any key information to objective or analysis to be addressed by the MSC database. I would suggest leaving only the most relevant databases, and includes the rest on a table indicating the country, region, time period covered, data source, type of data included, authors.

> **Authors Response:**
> We appreciate your comment. We have included a Table with the information that you suggest.

**Reviewer Specific Comment:**
6.- Page 4. Introduction Lines 24 to 30 I suggest to move to methodology section

    **Authors Response:**

    We appreciate your comment. We agree with what was suggested and we have relocated the paragraph that talks about the limitations and the bias derived from the application of the method in the methodology section.

**Reviewer specific Comment:**
7.- Page 4 introduction. Lines 31 to end of section, I suggest to delete this paragraph. Instead you should describe the specific objectives of this study.

    **Authors Response:**

    We agree with the suggestion and we have included a paragraph with the main objective and the sub-objectives of the work.

**Reviewer specific Comment:**
8.- Page 6, lines 13 to 15 probably not needed, delete.

    **Authors Response:**

    We have deleted this lines.

**Reviewer specific Comment:**
9.- Page 8. Indicate the list of damage types in a single line.

    **Authors Response:**

    We have indicated the list of damage types in a single line.

**Reviewer specific Comment:**
10.- Type of damages. Here roads and housing are the most common ones. I wonder if the news are bias to these two types because of most easy ones to be reported right after the event.

    **Authors Response:**

    The most frequent impacts caused by floods are usually road cuts. Riverbeds of the study area are of ephemeral functioning, therefore, a great part of them are crossed by the roads without bridges or, even, used as communication routes between the headwater and mouth areas. Therefore, it is logical that most of the damages are those of the roads. On the other hand, if we consider that floods are a natural risk, its measurement is based on the affection to human societies (Bates and Peacock, 1987, Tapsell, et al., 2002), therefore, it is not rare that another important part of the damages reports is housing. So, if we consider that the fact that road and housing damages are the most numerous, it does not imply a bias, but rather an evidence of the geographical and social reality of the floods in the study area. For clarity, we have added part of these arguments in section 4.4 (Flood damage variability) of the new version of the manuscript.

**Reviewer specific Comment:**
11.- page 10 line 25. How the quantity of damage was calculated?. In the case of housing, are you reporting the number of affected houses, or on roads, the number of cut roads...?

    **Authors Response:**

    The number of houses, the number of affected roads, and the number of other types of damage are not reported. For example, for each newspaper news about floods we assign, a value of 0 is assigned if there is no damage in a specific damage variable. Thus, this information only informs of absence or presence of damage, and not of the amount of each type of damage.

    In this respect, and as we pointed out in section 3 of the manuscript, damages are constructed as dichotomic variables to point to the presence (1) or absence (0) of any of the studied damages by flood in each municipality. Thus, information regarding the type of

damage suffered in each municipality was categorized in a simple way, it is, being aware of the difficulty involved in objectivizing quantitative information, consistently in time and space (Gil-Guirado et al., 2016).

**Reviewer specific Comment:**
12 Page 15. "cold drops" is a direct translation of the Spanish informal term. Please, use "cold pool" or mesoscale convective systems.

> **Authors Response:**
> Thanks for detecting the mistake. We have corrected this word.

**Reviewer specific Comment:**
13.- Page 15. From line 20 to 28, it is poorly written and they need major changes.

> **Authors Response:**
> We appreciate your comment. We agree with the suggestion and we will make major changes to correct that lines.

**Reviewer specific Comment:**
14. Page 17. Line 6. I don't understand "the latitudinal gradient referred to above continues to be reflected.

> **Authors Response:**
> We are sorry about the lack of clarity in this sentence. What we were trying to say is that the same latitudinal gradient is detected in that section (4.4 Flood damage variability) as the one mentioned in 4.2 Spatial Variability of floods.
> This latitudinal gradient is characterized by more severe, intensive, extensive and damaging floods as we move from north to south of the study area and it is mainly due to greater deficiencies in the spatial planning of the provinces in the south, although the climatic and orographic factors cannot be ruled out.
> In the new version of the manuscript, we have clarified line 6 on page 17, so that it is clear that we mean latitudinal gradient at this point.

**Reviewer specific Comment:**
15. Page 18. Line 25-27. This is not surprising due to the press nature of the database. As more small villages are cited on the newspaper, the flood extend on the database increases.

> **Authors Response:**
> As shown in panel b and c of figure 7, variability of annual cases of flooding and the annual area affected by floods seem very colinear. However, this should not necessarily be true, as there could be an increase in cases of flooding in municipalities with small size, while larger municipalities have a negative trend. In this way, panels b and c of figure 7 serve to confirm that there is no differential a flood trend in municipalities that has something to do with the surface of the municipalities. It is important to mention that, as indicated in section 4.2. of the manuscript, there is a great variability in the surface of the different municipalities studied.

**Reviewer specific Comment:**
16 Page 19. Lines 10-11. The sentence "The fact that the floods of L1 consider not only river floods (also consider flash floods and in situ floods), can magnify the importance of the increase in exposure, as to the growth of the exposed surface in flood zones." I wonder if the main problem is the nature of the database, because social perception of risk increase with time, since any single damage is reported on the local news.

> **Authors Response:**
> Again, as we pointed out in the response to your second general comment, in the text of the manuscript we mention the possible biases that risk perception can introduce (Llasat et al.,

2009) in the trends obtained. However, as we showed in that same response to your general comment, trends are mainly influenced by population growth and therefore, are influenced by the increased exposure to flood danger.

In any case, in the new version of the manuscript we have explained better the main idea in that paragraph.

**Reviewer specific Comment:**

17.-Conclusions should go to the point of the main results. In the present format, there are too long, and they should be shortened.

    **Authors Response:**

    Again, taking advantage of the deep revision of the language, we have proceeded to rewrite the conclusions so that they are more concise and focused on the results of the work. Thank you very much for your important contribution.

**Reviewer Other minor changes:**

Other minor changes are suggested on the pdf document: https://www.nat-hazards-earth-syst-sci-discuss.net/nhess-2019-10/nhess-2019-10-RC2-supplement.pdf

    **Authors Response:**

    We appreciate all your important suggestions. All minor changes has been taken into account in the new version of the document.

**Bibliography:**

Bates F. L. and W.G. Peacock (1987). Disasters and Social Change. In R.R. Dynes, B. Demarchi and C. Pelanda (eds.): *The Sociology of Disasters*. Franco Angeli Press, Milan

Clavero, B. S. (2016). Estadística de la cobertura de los riesgos extraordinarios en España por parte del Consorcio de Compensación de Seguros. Indice: *Revista de Estadística y Sociedad*, (67), 19-25.

Eisensee, T., & Strömberg, D. (2007). News droughts, news floods, and US disaster relief. *The Quarterly Journal of Economics*, *122*(2), 693-728.

Paprotny, D., Sebastian, A., Morales-Nápoles, O., & Jonkman, S. N. (2018). Trends in flood losses in Europe over the past 150 years. *Nature communications*, 9(1), 1985.

Tapsell, S. M., Penning-Rowsell, E. C., Tunstall, S. M., & Wilson, T. L. (2002). Vulnerability to flooding: health and social dimensions. *Philosophical transactions of the royal society of London. Series A: Mathematical, Physical and Engineering Sciences*, *360*(1796), 1511-1525.

Walmsley, D. J. (1980). Spatial bias in Australian news reporting. *Australian Geographer*, *14*(6), 342-349.

---

## Author Response (AR1)

Many thanks to the two referees and editor for their detailed and constructive reports.

We have made the changes in the original manuscript as reflected in the previous responses to the reviewers. In the revised version of the manuscript, all the changes in relation to the comments of the reviewers and editor are marked with active change control (the new text is marked in yellow).

5 Additionally, a thorough revision of the language has been carried out by a professional translator native in English. All these changes are also reflected with active change control

In summary, all the recommendations made by you and the reviewers have been considered, making the changes in the manuscript in relation to these recommendations and suggestions. In this way, we attach the revised manuscript with control changes to reflect the changes in relation to the comments of the

10 reviewers and editor.

Please, if you need any clarification, information or additional document, do not hesitate to contact us.

Below we reflect the specific changes in the original manuscript in relation to the suggestions of both reviewers:

**Referee #1 responses:**

15 **Overall comment:**

– **Reviewer general comment:** This is a very interesting paper, well-structured and written. The issues addressed are within the scope of NHESS. The construction of the SMC db is of high importance for the analysis of the spatial and temporal changes in the coastal areas vulnerability to floods. Conclusions can be very useful to decision-makers for adaptation planning. The methodology followed for the development of the SMC database is appropriate and well presented.

20 Therefore, the article merits being published, with minor changes.

– **Authors response:** Thank you for your flattering and constructive comments. We believe that the results of this work may represent an improvement in the knowledge of the spatio-temporal patterns of floods in the Spanish Mediterranean coast in the last 50 years. We have included all your suggestions in the new manuscript version, this will make the manuscript undoubtedly more robust.

25 **Specific comments:**

– **Reviewer comment:** 1. My only scientific concern is the use of 2 different averaged indices for the average impact severity. I mean, the intensity level is actually related to the impact magnitude: low damages / major / deaths and/or general destruction. Then the authors produce a damage severity index, which uses in the equation the intensity level as weight implemented on the various damages occurrence. I can understand that cases can be compared better based on the

30 severity index. However, I am not convinced about the use of 2 'average' values used to evaluate trends or vulnerability at aggregated areas. What is the point? Maybe this could be better explained.

- **Authors response:** We agree with you that the use of two different means can lead to confusion. However, we believe that the comparison between these two indices can help to have a better idea of the problems generated by floods in the study area. We consider that the average intensity is a robust measure, since it limits the accumulated bias of adding several quantitative magnitudes extracted from qualitative information. In addition, the average intensity is based on numerous works that use the three levels of intensity for the study of floods through historical and hemerographic documentation (Camuffo and Enzi, 1996; Barriendos et al., 2003; Llasat, et al., 2005; Barriendos et al., 2014). However, we believe that the severity index provides additional information, although it may be subject to greater subjectivity. As an example, a flood of intensity 2 could produce major damage, but with concentrated effects in agriculture (Intensity = 2, severity index = 2), while a flood of intensity 1, could have some weak damages, but extended to a large number of sectors (for example roads, tourism, commerce and agriculture) (Intensity = 1; severity index = 4 * 1 = 4). In other cases a flood could be very intense and also affect a large number of sectors, so its final impact is greater than if simply considering the intensity (For example, a flood of intensity 2, which affected roads, agriculture, tourism and trade would have a severity index of 8). Therefore, the severity index offers information that is complementary to the intensity level and the amount of damages.
  - **Manuscript changes:** we have introduced part of the previous discussion in the new version of the manuscript (pages 8 and 9). Additionally, we have moved the part of section 4 (Results) where the calculation of additional variables was discussed, to section 3 (Methodology and sources).

- **Reviewer comment:** 2. In what concerns the structure of the paper, my only concern is the introduction. In page 4, paragraph 4 (lines 24-29) is too methodological to be included in the introduction. It confuses the reader who expects to read the objectives and research questions instead of fragmentary information about methods employed. I suggest this part to be transferred to the methods section.
  - **Authors response and Manuscript changes:** we agree with you that this part of the work is not appropriate for the introduction section and that it is more typical of the methodology section. However, in order to not repeat information in the methodology section, we have delete most of the text from lines 24 to 29 on page 4 in the new version of the manuscript.

- **Reviewer comment:** 3. Please consider for your references regarding the databases in other countries also the high-impact weather events database of the National Observatory of Athens, Greece, which is also active on-line, constantly updated and with weather and impact intensity classification (10.5194/nhess-13-727-2013). The NOA db has been also based on press articles.
  - **Authors response and Manuscript changes:** Thank you very much for your important suggestions. We have included the reference 10.5194 / nhess-13-727-2013 in the introduction section and we have pointed out the peculiarities of this database. Additionally, we have added its hemerographic characterization to the description of the NOA database (See lines 26 to 28 in page 2 in the new manuscript version).

**Technical corrections:**

- **Authors response:** Thank you very much for highlighting these important details and providing advice about the convenience of including some necessary clarifications. All these issues have been taken into account. Most of the corrections have been resolved thanks to a native English speaker who will be responsible for reviewing the text of the new version before being sent.
  Here is a more detailed description of the changes carried out:

- **Reviewer comment:**
  1. P3, l15: Please delete the 'y'.

– **Manuscript changes:** Delete

2. P4, l10: Please delete 'but'.

   – **Manuscript changes:** Delete

3. P4, l14: It is 'flood cases', not 'floods cases'. Please repeat correction throughout the article.

   – **Manuscript changes:** this mistake has been corrected

4. P4, l15: Please correct as: In this regard it must be clarified the difference between flood cases and flood events

   – **Manuscript changes:** this mistake has been corrected

5. P4, l24: Please add 'digital' before 'archives'.

   – **Manuscript changes:** word included

6. P4, l29: The sentence is too big. Please start a new one from 'in general.

   – **Manuscript changes:** we have corrected this sentence in the revised manuscript.

7. P4, l33: Please delete 'This is', otherwise the sentence does not make sense. In the same sentence, please use the same term throughout the paper regarding the flood 'case'. You have explained very well in the document the difference between case and event. So, the words 'episodes' and 'events' in this sentence do not fit.

   – **Manuscript changes:** Thank you very much for your corrections. We have considered your suggestions and have replaced events for cases.

8. P5, l17: Please correct as 'emphasized'. Also, please rephrase the entire sentence as it is not clear, especially the second part.

   – **Manuscript changes:** we have corrected and revised this sentence in the revised manuscript.

9. P6, l4: Please explain the: (2003: 800)

   – **Manuscript changes:** we have deleted ":800".

10. P6, l5: please correct as: These situations.

   – **Manuscript changes:** this mistake has been corrected

11. P6, l10: please use the same term: environmental or climatic

   – **Manuscript changes:** we have replaced environmental by climatic

12. P6, l13: just a thought: is this sentence for Franco necessary?

   – **Manuscript changes:** probably this sentence is not necessary. So, we have deleted this sentence.

13. P6, l18: please cut this sentence in 2 parts. it is too big and difficult to read

- **Manuscript changes:** we have corrected this paragraph and shortened the sentences in the revised manuscript.

14. P6, l19: it is weird the use of 'it has:' after 'the following criteria'. I think it can be improved.

- **Manuscript changes:** we have corrected this sentence in the revised manuscript.

15. Table 1: what is MEDIFLOOD? do you mean SMC-Flood db? Also, the authors could enter an extra column to report the cities of head offices. The full newspaper names could be added here as a comment.

- **Manuscript changes:** we have corrected the mistake and replaced MEDIFLOOD by SMC-Floods database. We have also added an extra column in the table to report the head offices cities. Additionally, the full newspaper names are added as a footnote to the table.

16. P6, last paragraph: the different names are confusing. Consider keeping the short names of Table 1 everywhere in the text.

- **Manuscript changes:** We have considered your suggestion and for the sake of clarity, we have kept the short names of Table 1 throughout the text.

17. P7, l5: 1) the sentence is too big. Please enter full-stop before 'Taking into account'. 2) Please consider avoiding the footnote since it concerns only one source. You could include it in the text instead.

- **Manuscript changes:** we have corrected this paragraph and shortened the sentences in the revised manuscript. Additionally, we have added the reference of note 1 in a sentence within the main text.

18. P7, l7: correct as 'validated'

- **Manuscript changes:** this mistake has been corrected

19. P7, l8: please delete 'de'. L10: please add space before 'Secondly'. L22: please correct as 'within'

- **Manuscript changes:** all this mistakes has been corrected

20. P10, l4: Please begin a new sentence at 'Multiplying:

- **Manuscript changes:** we have corrected this paragraph and shortened the sentences in the revised manuscript.

21. Table 2: maybe it is better if you write 'average severity index'

- **Manuscript changes:** we have added this suggestion

22. P11, l7: Please add 'the' before 'number'

– **Manuscript changes:** "the" has been added

23. P11, l8: please consider defining 'average intensity', as this is the first time we read this.

– **Manuscript changes:** following this suggestion, the definition of average intensity has been added to this paragraph (See lines 5 to 6 in page 10 in the new manuscript version)

24. P15, l5: 'concentration' of what? I think something is missing

– **Manuscript changes:** Thanks for detecting this mistake. In this sentence we want to show that during the fall there is a greater concentration of floods as intensity increases. Therefore we have modified the phrase to include the word floods: "during this season there is a higher concentration of floods as the intensity increases".

25. P19, l11: Please add 'to' before 'add'

– **Manuscript changes:** "to" has been added

26. P20, l3: please correct as 'makes us'

– **Manuscript changes:** this mistake has been corrected

**Referee #2 responses:**

**Overall comment:**

– **Reviewer General Comment:** The paper "SMC-Floods database: A high resolution press database on floods for the Spanish Mediterranean Coast (1960-2015)" provides a preliminary description and analysis of flood data collected from press news. It is not a novel initiative at European or Spanish level, but it comprises a large extension of a flood damage prone region. I admit that such effort merits publication somewhere, but not in the present format, which requires a major review before it can be published.

– **Authors Response:** Thank you for your suggestions, we have included all of them in the new manuscript version. The focus of our work is to cover two main needs:

1. Analyze the trends of flood cases and events in the Spanish Mediterranean coast. This is an area that has increased the number of floods and, according to the IPCC (2012), there are great uncertainties about the importance of the physical factor and the human factor in the balance of economic losses caused by floods.

2. The final goal of this study is focused on knowing to what extent the variability of floods is caused by changes in the social systems. In this regard, we consider that other floods databases show a lack of data of floods in small towns (Paprotny et al., 2018) and we show that the lack of information has a substantial impact on observed trends.

– **Reviewer General Comment:** In scientific outcomes are highly bias by the journalist judgment of the flood damages and newspaper coverage and audience. Therefore, caution should be placed on the interpretation of the data.

- **Authors response and manuscript changes:** we share your view of this point, for this reason the manuscript included a paragraph of assumption of limitations that takes into account the possible bias indicated by the reviewer: "*However, there are other factors that should not be overlooked or dismissed. According to Llasat et al., 2009, it is important to consider that trends may be biased by various reasons: i) a greater sensitivity or perception towards natural risks from the public opinion could increase news about floods in newspapers; ii) a greater spatial coverage of the news thanks to the improvement of communications. The complexity of the factors (social, cultural, environmental and perceptual) involved in flood processes make us think about the influence of the mentioned items in the observed trends. Therefore, a deeper knowledge on climatic, geographic and socioeconomic variables involved is necessary*".

To the above explanation we have added another possible biased commentary to the manuscript text: Eisensee and Strömberg (2007) argue that the coverage of natural disasters in the press depends on the availability of other newsworthy material at the time of the disaster. Additionally, and as you point out, there may be some spatial bias in the news based on the newspaper's spatial coverage (Walmsley, 1980). In this regard, and as explained in section 3 (Methodology and sources), the newspapers used are regional newspapers, which specifically cover the information of each of the Autonomous Communities studied throughout the study period. On the other hand, the experience of other related works (Llasat et al., 2009) and the way of gathering information, show that the subjectivity of the journalist or the newspaper can bias the level of intensity, but not the type of damage. In this regard, the type of damage may be under-documented, but it can rarely be over-documented. However, it is true that the increase of population has been able to influence the increase of news in small populations. This is precisely one of the facts that we highlight in this work, since a large part of the detected trends are influenced by the population increase. This fact can be observed when we analyze the trends in floods according to population growth. In making this analysis, we observed that the greater the population increase of the municipalities between 1960 and 2011 was (the two extreme census moments during the study period), the greater significance and intensity the trend detected has. In the following table (Table 1) it can be observed that in the set of municipalities where population have grown less than 50% between 1960 and 2011, the floods have no significant trend. However, floods have a statistically significant trend in municipalities that have grown more than 50%. The interesting thing is that the rate of increase in floods is greater as the population growth is greater. This shows that the detected trends are largely influenced by the increase in exposure.

Table 1: Floods trends in Spanish Mediterranean Coastal Municipalities related to ranges of population increase between 1960 and 2011.

| Population increase range in % | Kendall's tau | P-value | Sen´s Slope |
|---|---|---|---|
| Less than 0% | -0.048 | 0.733 | 0 |
| Between 0 and 50% | -0.060 | 0.550 | 0 |
| More than 50 and less than 100% | 0.280 | 0.005 | 0.127 |
| More than 100 and less than 200% | 0.340 | 0.000 | 0.286 |
| More than 200 % | 0.380 | < 0,0001 | 0.471 |

*To calculate trends, we have used Hirsch and Slack's nonparametric test (1984), which is based on Mann-Kendall range. The trial version of XLSTAT software (Addinsoft, 2018) was used to calculate it. The Mann-Kendall test provides a level of statistical significance (p-value). The threshold of significance chosen was 95%, which indicates that p-values above 0.05 should lead to rejecting the hypothesis of a trend in the series. When the p-value is less than 0.05, the trend can be positive or negative. Sen´s Slope shows the annual change rate in floods. That is, the value informs about the annual increase or decrease of the floods.*

To clarify this point, we have added the previous table and part of the previous comments to section 4.5. in the new manuscript version (See lines 2 to 30/page 20). However, it is true that there are still many uncertainties regarding the possible information bias. For this reason, in the conclusions section we have included that, although the results are robust when relating population increase and increase in floods, the trends detected may be biased by journalistic issues (See lines 26 to 29/page 21).

- **Reviewer General Comment:** there is not a critical analysis of the results in relation to other more robust database, for instance the analysis from the National Insurance Consortium. As this database reflects risks (mostly exposure and vulnerability), most of the hydroclimatologic trends and changes on hydroclimatic conditions may not be valid.
  - **Authors response and manuscript changes:** we agree with the reviewer. In the new version of the manuscript, a critical analysis is included in relation to the National Insurance Consortium database (See lines 12 to 27/page 18). However, it should be noted that the Spanish insurance contract law does not require the insurance of the home. Therefore, this database could be even more biased than ours depending on the degree of insurance coverage in the municipalities of the study area (Clavero, 2016). On the other hand, the fact that the National Insurance Consortium database is based on private insured assets limits information on the impact of floods on public goods such as roads. In this regard, and taking into account the great weight that road damages have on our database, it is not surprising that the SMC-Floods database considers a number of cases far superior to the National Insurance Consortium database.

- **Reviewer Specific Comment:** 1.- The manuscript requires a detail English correction on the style. It looks a direct translation from a Spanish text, I would say that the authors used google translator, otherwise, I cannot explain the use of some very incorrect terms. Among the most critical one are "Cold Drop" cited in the paper, and probably authors refer to "cold pool" or "mean mobile" (cited in figure 7) instead of "moving average". These are only few examples, but the text is full of informal terms or sentences that do not make any sense in English.
  - **Authors response and manuscript changes:** a native English speaker has been responsible for reviewing in depth the text of the new version of the manuscript.

- **Reviewer Specific Comment:** 2.- The manuscript is very long and this makes difficult to read. The authors should analyze in each sentence and use proper language addressing the point in a direct way.
  - **Authors response and manuscript changes:** the deep revision of the language of the manuscript by a native English speaker, has been an important summary and synthesis of the manuscript. In this way, we consider that the ideas are now expressed more clearly.

- **Reviewer Specific Comment:** 3.- Several sections can be shortened, including the introduction and conclusions.
  - **Authors response and manuscript changes:** the same than the last point. Additionally, we have eliminated non-relevant text parts in the introduction and conclusions sections.

- **Reviewer Specific Comment:** 4.- Abstract: The abstract should be completely re-written. The way it is written looks and introduction rather than a summary. Sentences such as "Floods are the natural disaster that affects the greatest number of people and causes the highest economic losses in the world" are fine for the introduction, but not for the abstract. Please, start the abstract by telling the reader at once what the paper is: new data, a review of progress, a new technique, a synthesis, or whatever describes the nature of the paper. Unnecessary descriptive phrases and qualifiers should be left out of the abstract. Write the abstract as styled summary of its essential information; and include as much specific information as possible on the results.
  - **Authors response and manuscript changes:** we appreciate the comment of the reviewer and we have rewritten completely the abstract following his suggestions.

- **Reviewer Specific Comment:** 5.- Introduction: There is a long description of flood databases from press news in Europe and the world, and they do not provide any key information to objective or analysis to be addressed by the MSC database. I would suggest leaving only the most relevant databases, and includes the rest on a table indicating the country, region, time period covered, data source, type of data included, authors.
  - **Authors response and manuscript changes:** We appreciate your comment. Following your recommendation in comment 3, we have shortened the Introduction section. For this, we have chosen to describe the most relevant databases for the objectives of this work and for more general bases, we have chosen to refer to previous works that describe these databases in tables and graphs.

- **Reviewer Specific Comment:** 6.- Page 4. Introduction Lines 24 to 30 I suggest to move to methodology section
  - **Authors response and manuscript changes:** we agree with you that this part of the work is not appropriate for the introduction section and that it is more typical of the methodology section. However, in order to not repeat information in the methodology section, we have delete most of the text from lines 24 to 29 on page 4 in the new version of the manuscript.

- **Reviewer specific Comment:** 7.- Page 4 introduction. Lines 31 to end of section, I suggest to delete this paragraph. Instead you should describe the specific objectives of this study.
  - **Authors response and manuscript changes:** we agree with the suggestion and we have included a paragraph with the main objective and the sub-objectives of the work. Además, hemos eliminado las líneas 31 al final de la sección en la página 4.

- **Reviewer specific Comment:** 8.- Page 6, lines 13 to 15 probably not needed, delete.
  - **Authors response and manuscript changes:** we have deleted this lines.

- **Reviewer specific Comment:** 9.- Page 8. Indicate the list of damage types in a single line.
  - **Authors response and manuscript changes:** we have indicated the list of damage types in a single line.

- **Reviewer specific Comment:** 10.- Type of damages. Here roads and housing are the most common ones. I wonder if the news are bias to these two types because of most easy ones to be reported right after the event.
  - **Authors response and manuscript changes:** the most frequent impacts caused by floods are usually road cuts. Riverbeds of the study area are of ephemeral functioning, therefore, a great part of them are crossed by the roads without bridges or, even, used as communication routes between the headwater and mouth areas. Therefore, it is logical that most of the damages are those of the roads. On the other hand, if we consider that floods are a natural risk, its measurement is based on the affection to human societies (Bates and Peacock, 1987, Tapsell, et al., 2002), therefore, it is not rare that another important part of the damages reports is housing. So, if we consider that the fact that road and housing damages are the most numerous, it does not imply a bias, but rather an evidence of the

geographical and social reality of the floods in the study area. For clarity, we have added part of these arguments in section 4.4 (Flood damage variability) of the new version of the manuscript (see lines 12 to 16/page18).

– **Reviewer specific Comment:** 11.- page 10 line 25. How the quantity of damage was calculated?. In the case of housing, are you reporting the number of affected houses, or on roads, the number of cut roads...?
  – **Authors response and manuscript changes:** the number of houses, the number of affected roads, and the number of other types of damage are not reported. For example, for each newspaper news about floods we assign, a value of 0 is assigned if there is no damage in a specific damage variable. Thus, this information only informs of absence or presence of damage, and not of the amount of each type of damage.
  In this respect, and as we pointed out in section 3 of the manuscript, damages are constructed as dichotomic
  variables to point to the presence (1) or absence (0) of any of the studied damages by flood in each municipality.
  Thus, information regarding the type of damage suffered in each municipality was categorized in a simple way,
  it is, being aware of the difficulty involved in objectivizing quantitative information, consistently in time and
  space (Gil-Guirado et al., 2016).

– **Reviewer specific Comment:** 12 Page 15. "cold drops" is a direct translation of the Spanish informal term. Please, use "cold pool" or mesoscale convective systems.
  – **Authors response and manuscript changes:** thanks for detecting the mistake. We have corrected this word.

– **Reviewer specific Comment:** 13.- Page 15. From line 20 to 28, it is poorly written and they need major changes.
  – **Authors response and manuscript changes:** we appreciate your comment. We agree with the suggestion and we have made major changes to correct that lines.

– **Reviewer specific Comment:** 14. Page 17. Line 6. I don't understand "the latitudinal gradient referred to above continues to be reflected.
  – **Authors response and manuscript changes:** we are sorry about the lack of clarity in this sentence. What we were trying to say is that the same latitudinal gradient is detected in that section (4.4 Flood damage variability) as the one mentioned in 4.2 Spatial Variability of floods.
  This latitudinal gradient is characterized by more severe, intensive, extensive and damaging floods as we move
  from north to south of the study area and it is mainly due to greater deficiencies in the spatial planning of the
  provinces in the south, although the climatic and orographic factors cannot be ruled out. In the new version of
  the manuscript, we have clarified line 6 on page 17, so that it is clear that we mean latitudinal gradient at this
  point.

– **Reviewer specific Comment:** 15. Page 18. Line 25-27. This is not surprising due to the press nature of the database. As more small villages are cited on the newspaper, the flood extend on the database increases.
  – **Authors response and manuscript changes:** as shown in panel b and c of figure 7, variability of annual cases of flooding and the annual area affected by floods seem very collinear. However, this should not necessarily be true, as there could be an increase in cases of flooding in municipalities with small size, while larger municipalities have a negative trend. In this way, panels b and c of figure 7 serve to confirm that there is no differential a flood trend in municipalities that has something to do with the surface of the municipalities. It is important to mention that, as indicated in section 4.2. of the manuscript, there is a great variability in the surface of the different municipalities studied.

- **Reviewer specific Comment:** 16 Page 19. Lines 10-11. The sentence "The fact that the floods of L1 consider not only river floods (also consider flash floods and in situ floods), can magnify the importance of the increase in exposure, as to the growth of the exposed surface in flood zones." I wonder if the main problem is the nature of the database, because social perception of risk increase with time, since any single damage is reported on the local news.
  - **Authors response and manuscript changes:** as we pointed out in the response to your second general comment, in the text of the manuscript we mention the possible biases that risk perception can introduce (Llasat et al., 2009) in the trends obtained. However, as we showed in that same response to your general comment, trends are mainly influenced by population growth and therefore, are influenced by the increased exposure to flood hazard. In any case, in the new version of the manuscript we have explained better the main idea in that paragraph.

- **Reviewer specific Comment:** 17.-Conclusions should go to the point of the main results. In the present format, there are too long, and they should be shortened.
  - **Authors response and manuscript changes:** taking advantage of the deep revision of the language, we have proceeded to rewrite the conclusions so that they are more concise and focused on the results of the work. Thank you very much for your important contribution.

- **Reviewer Other minor changes:** Other minor changes are suggested on the pdf document: https://www.nat-hazards-earth-syst-sci-discuss.net/nhess-2019-10/nhess-2019-10-RC2-supplement.pdf
  - **Authors response and manuscript changes:** we appreciate all your important suggestions. All minor changes has been taken into account in the new version of the document.

[revised manuscript text omitted]

---

## Referee Report (RR1)

**The article merits to be published after the implemented changes and corrections. However, I have a specific comment about the abstract, which is the 'front' page and has to be correct, precise and attractive.**

**Specific comment:**

To my experience the phrase 'High spatiotemporal database' is not appropriate and does not mean anything. The authors probably mean High spatiotemporal resolution flood database or Flood database of high spatiotemporal resolution.

Please consider revising.

Thank you for your efforts.

---

## Author Response (AR2)

**Editor Decision: Publish subject to technical corrections (12 Aug 2019) by Maria-Carmen Llasat**

Comments to the Author:

5   Dear Drs. Gil-Guirado, Pérez-Morales, Lopez-Martinez
Thank you very much for your interesting paper. As you know, two reviewers have now provided detailed reviews of your revised version. It is a pleasure for me telling you that both have both referees coincide in their dictamen and accept to publish your paper after some technical corrections.
Referee 1 tell you that the phrase "'High spatiotemporal database' is not appropriate and does not mean anything."
10  I haven't found this expression and you can maintain high resolution database. The second referee proposes some little changes that you should introduce in the definitive version.
Line 29, page 10. The sentence "This calculation informs that, in the SMCM, each flood affects 30 an area of 119 km2" is not correct. You can say the "mean municipal area affected by a flood case is 119km2". In consequence, modify also line 25 page 18. Besides this, in my last report I have asked you for introducing the expression flood
15  case in all the parts of the text in which you are referring the flood affectation in a municipality. However, they are still some sentences in the Results in which you have don't do it, not in the text neither in the figures (i.e. Figure 7 axe y is flood cases not flood, and the same in the text in the foot of this figure). Please, review with accuracy the document and substitute flood by flood case when necessary. This is particularly important in the part devoted to trend analysis and spatial analysis. Why is this question so much important? Because readers can reproduce your
20  words/information and propagating a nonsense data. As example, following Figure 7, in 1989 were produced 200 floods in the Spanish Mediterranean Region (the correct text would be 200 flood cases").
I look forward to seeing the next version of your manuscript that I hope will be able to be published directly.
Regards
María Carmen Llasat
25  NHESS Editor
Full Professor of Atmospheric Physics, University of Barcelona

**Authors' response:**
We are very happy to know that the paper will be accepted for publication in NHESS. Thank you very much for
30  your work and kind help.
Thank you very much for detecting all the errors, and especially, thanks for pointing out the need to clearly differentiate between "floods" and "flood cases". As you point out, these differences must be clarified, so that errors or confusions do not occur. In this regard, we have replaced "floods" by "flood cases" both in figures 4 and 7, as in the title and the subtitles of the manuscript, as well as in the whole document. All these changes are marked with
35  control change and highlighted in yellow.
In the same way, we have followed all the recommendations and changes indicated by both referees, and we have proceeded to make the changes in the document (these changes are also marked with control change and highlighted in yellow).
Again, thank you very much for your work and help.
40  We are available for anything else that needs to be done.
Regards

**Referee 1#:**
The article merits to be published after the implemented changes and corrections. However, I have a specific
45  comment about the abstract, which is the 'front' page and has to be correct, precise and attractive. Specific comment: To my experience the phrase 'High spatiotemporal database' is not appropriate and does not mean

anything. The authors probably mean High spatiotemporal resolution flood database or Flood database of high spatiotemporal resolution. Please consider revising. Thank you for your efforts.

- − **Author s' response:** siguiendo las indicaciones del editor y las del referee 1#, hemos sustituido "High spatiotemporal database" por "Flood database of high spatiotemporal resolution"

**Referee 2#:**

The paper has been highly improved in this second version, and it is ready for publication after some minor changes.
Page 1. Abstract. Line 1. Consider start as follow:
Long-term flood databases covering large regions are a necessary tool….

- − **Author s' response:** following the recommendations of the editor and referee 1, the abstract now start as follow: "Flood database of high spatiotemporal resolution are a necessary tool for proper spatial planning, especially in areas with high levels of exposure and danger to floods".

Consider to change the last sentence to: This pattern calls for new actions by the coastal municipal authorities of southern Spain for adaptation to a more complex flood scenario.

- − **Author s' response:** following its recommendation we have replaced the phrase "This pattern subjects the coastal municipalities of the south of Spain to a complex floods adaptation scenario", by this other: "This pattern calls for new actions by the coastal municipal authorities of southern Spain for adaptation to a more complex flood scenario".

Line 23. In singular: "society"
Line 24-25: This extreme hydroclimatic behavior has exposed society, who need water resources for their agricultural and domestic demands, to large damages by torrential nature of rainfall (ref)
Line 1-2: ….. human intervention that has taken…
Line 10. Furthermore, this climate conditions can become….
Note that the word scenario implies a number of assumptions in a model initial conditions, but in your case you are talking about real climate observations, so this is not an scenario but natural climate conditions
Line 13. Added to this climate conditions is the effect..
Line 18. In addition, this extreme climate is complicated..
Page 17. Line 8 should be carried on following the previous sentence at line 7.
Page 18. Line 3 should be carried on following the previous sentence.

- − **Author s' response:** following their recommendations, all previous changes have been made (all changes are marked with control change and highlighted in yellow).

[revised manuscript text omitted]